# LoRAtorio: An intrinsic approach to LoRA Skill Composition

## Abstract

Low-Rank Adaptation (LoRA) has become a widely adopted technique in text-to-image diffusion models, enabling the personalisation of visual concepts such as characters, styles, and objects. However, existing approaches struggle to effectively compose multiple LoRA adapters, particularly in open-ended settings where the number and nature of required skills are not known in advance. In this work, we present LoRAtorio, a novel train-free framework for multi-LoRA composition that leverages intrinsic model behaviour. Our method is motivated by two key observations: (1) LoRA adapters trained on narrow domains produce unconditioned denoised outputs that diverge from the base model, and (2) when conditioned out-of-distribution, LoRA outputs show behaviour closer to the base model than when conditioned in distribution. In the single LoRA scenario, personalisation and customisation show exceptional performance without catastrophic forgetting; the performance, however, deteriorates quickly as multiple adapters are loaded. Our method operates in the latent space by dividing it into spatial patches and computing cosine similarity between each patch's predicted noise and that of the base model. These similarities are used to construct a spatially-aware weight matrix, which guides a weighted aggregation of LoRA outputs. To address domain drift, we further propose a modification to classifier-free guidance that incorporates the base model's unconditional score into the composition. We extend this formulation to a dynamic module selection setting, enabling inference-time selection of relevant LoRA adapters from a large pool. LoRAtorio achieves state-of-the-art performance, showing up to a 1.3% improvement in CLIPScore and a 72.43% win rate in GPT-4V pairwise evaluations, and generalises effectively to multiple latent diffusion models. Code will be made available.

## 1 Introduction

Diffusion models operate by gradually learning to reverse a noise process, effectively capturing the underlying data distribution through iterative denoising (Ho et al., 2020; Nichol & Dhariwal, 2021; Song et al., 2021). In practice, this enables them to approximate the complex structure of their training data and generate new, previously unseen samples that remain faithful to the original data's domain. Beyond base text-to-image generation capabilities, works such as Dreambooth (Ruiz et al., 2023) and StyleDrop (Sohn et al., 2023) have enabled personalisation and fine-grained customisation. These approaches often rely on LoRA adapters (Hu et al., 2022), which specialise a base model to preserve the identity of specific concepts or objects, supporting applications like virtual try-on (Lobba et al., 2025) and avatar generation (Huang et al., 2024b). Each LoRA adapter effectively encodes a "skill" or concept, and generation with a single adapter yields precise, high-quality outputs. However, when multiple skills are loaded simultaneously into a single model instance, we observe a rapid deterioration in performance (Zhong et al., 2024; Prabhakar et al., 2025). Understanding the source of this degradation is key to enabling reliable multi-concept generation.

To better understand the challenges of composing multiple LoRA adapters, we begin with a preliminary analysis of their behaviour. Specifically, we examine the unconditional noise representations produced by the base model and various LoRA-augmented models. We observe that the distribution of the LoRA diverges from that of the base model (Figure 1), particularly when LoRAs are trained on narrow or highly specialised datasets—conditions common in personalisation settings (Li et al., 2024b; Ruiz et al., 2023). This domain shift also manifests in conditioned outputs, as evident through visual inspection (Figure 2).

**Observation 1** *The unconditioned noise estimate $e_i(z, t)$ produced by the $i^{th}$ LoRA differs from that of the base model $e(z, t)$.*

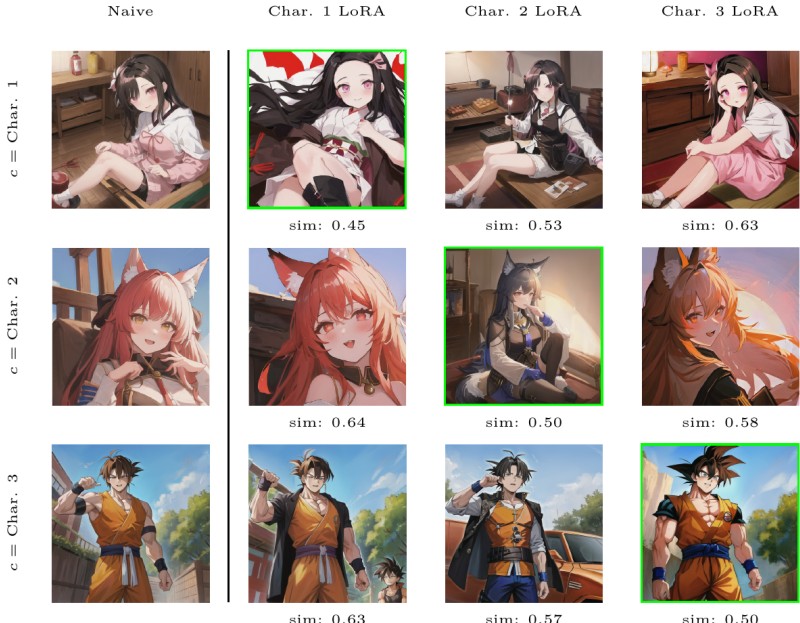

Figure 2: Generated images from each Character LoRA, conditioned with text prompts originally associated with other LoRAs in the *ComposLoRA* testbed. When a prompt falls outside a LoRA's training distribution, the predicted latent $e_{\theta_i}(z_0, 0, c)$ of the $i^{th}$ LoRA tends to align closely with that of the base (Naïve) model, showing minimal deviation due to changes in $p_{\theta_i}(x)$, also shown by the cosine similarity of the conditioned latent of the Naïve model $\tilde{e}_\theta(z_0, 0, c)$ with that of the $i^{th}$ LoRA $\tilde{e}_{\theta_i}(z_0, 0, c)$.

While this divergence is notable, especially under unconditional or in-domain conditions, LoRA adapters are also known to mitigate catastrophic forgetting, particularly compared to fully fine-tuned models (Biderman et al., 2024). That is, LoRAs tend to preserve the base model's generalisation capabilities. Indeed, we observe that even though there are stylistic changes in the generated image as a result of the loaded LoRAs, the composition and theme of the generated output more closely resemble the base model when a text condition outside the LoRA distribution is given. This is attributed to the sparse and low-norm nature of LoRA weights (Fu et al., 2023; Shah et al., 2024). To quantify this effect, we measure cosine similarities between the noise scores of LoRA-augmented models and the base model, both within and outside LoRA's training distribution. These measurements, along with visual inspection (Figure 2), support the following:

**Observation 2** *When the input condition lies outside the LoRA's target domain, the output of the augmented model more closely resembles that of the base model.*

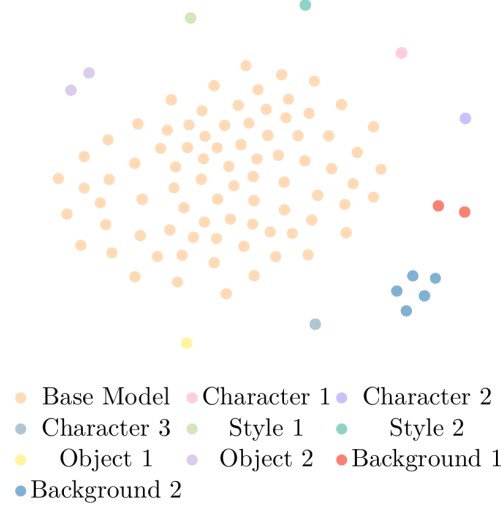

Figure 1: t-SNE visualisation of the unconditioned latent space, for the Base Model and LoRA-adapted models in *ComposLoRA* testbed.

Previous works in skill composition for image generation have used both trainable (Charakorn et al., 2025; Shenaj et al., 2024; Zhu et al., 2024) and train-free approaches (Zhong et al., 2024; Zou et al., 2025; Li et al., 2024a; Yang et al., 2024). The former have focused on either trainable mixture-of-experts (Zhu et al., 2024) or training a hyper-network that generates LoRA weights of the combined task (Shenaj et al., 2024). However, trainable methods are impractical in real-life applications as they would require re-training for every

new concept or domain added. Furthermore, in several commercial applications, training data may not be available due to confidentiality constraints, thereby raising the need for train-free skill composition. Inference time composition in image generation is relatively unexplored, with methods focusing on a schedule of LoRAs based on prior knowledge (Zhong et al., 2024; Zou et al., 2025), additional conditions (Yang et al., 2024) or merging of latent space (Zhong et al., 2024), breaking away from weight manipulations (Hugging Face, 2024; Huang et al., 2024a; Shah et al., 2024; Li et al., 2024a), which can show diminished performance as the number of skills incorporated increases. However, unweighted merging of scores will eventually face similar issues to weight merging and setting a schedule requires prior knowledge of the task at hand. However, all previous approaches assume that the set of LoRAs to be composed is known in advance and manually specified by the user. In practice, this assumption rarely holds. Real-world applications such as personalised advertising or interactive content generation often require adapting to user intent or contextual cues that are only available at inference time. In such scenarios, pre-selecting or pre-scheduling LoRAs becomes impractical—both because the relevant concepts may not be known beforehand, and because the combinatorial space of possible LoRA mixtures grows rapidly with the number of skills.

In this work, motivated by Observation 1 and Observation 2, we propose LoRAtorio, a train-free method for multi-LoRA skill composition in image generation. Our approach leverages the intrinsic behaviour of LoRA-augmented models without requiring additional supervision or fine-tuning. Specifically, we introduce a fine-grained mechanism that operates in the latent space by dividing it into spatial patches. For each patch, we compute the cosine similarity between the output of the LoRA-augmented model and that of the base model. These similarities are used to construct a spatially-aware weight matrix, where patches that deviate more from the base model receive higher weights. This matrix is then used to compute a weighted average of the predicted noise outputs across LoRAs, allowing the model to emphasise regions where individual LoRAs are more confident. To mitigate domain drift, we propose a modification to the classifier-free guidance mechanism by incorporating the base model's unconditional noise estimate into the weighted average. This adjustment ensures that the final output remains grounded in the base model's general knowledge. Unlike prior approaches that rely on extrinsic signals such as frequency (Zou et al., 2025) or empirical scheduling (Zhong et al., 2024), LoRAtorio is entirely based on intrinsic model behaviour–specifically, the consistency between LoRA and base model representations. Finally, we extend the task to a dynamic module selection setting, in which all available LoRA adapters are loaded into the base model, and the most relevant ones are selected ad hoc during inference. This formulation more realistically reflects real-world skill composition scenarios, where the set of required capabilities is not known a priori.

Our main contributions can be summarised as follows:

- We introduce **LoRAtorio**, a train-free and intrinsically guided approach for multi-LoRA composition in diffusion models, leveraging spatially-aware similarity to the base model.

- Furthermore, we propose re-centering the unconditioned score in classifier-free guidance to address domain drift caused by personalisation training.

- We extend the task of multi-LoRA composition to a dynamic module selection setting, where all LoRA adapters are loaded into the base model and selected at inference time based on intrinsic similarity.

We demonstrate that LoRAtorio achieves state-of-the-art (SoTA) performance on the *ComposLoRA* benchmark both in terms of automated metrics and human preference. This is consistent for both static and dynamic module settings. Furthermore, we extend our evaluation to a rectified flow (Esser et al., 2024) architecture, showing our method's robustness.

## 2 LoRAtorio

**Preliminaries:** Latent Diffusion Models (Rombach et al., 2022) operate by performing the denoising diffusion process in a learned latent space. Given an input $x_0$, an encoder $\mathcal{E}$ maps it to a latent representation $z_0 = \mathcal{E}(x_0)$; during the diffusion process, Gaussian noise is progressively added to $z_0$, thus producing a noisy sequence $\{z_t\}_{t=1}^T$. The diffusion model learns to approximate the reverse process via a denoising network $e_\theta(z_t, t, c)$, conditioned on context $c$. Classifier-free guidance (CFG) (Ho & Salimans, 2021) is incorporated by training the model with both conditional and unconditional objectives. During sampling, guidance is applied by modifying the predicted noise as follows:

$$\hat{e}_\theta(z_t, t, c) = e_\theta(z_t, t) + s \cdot (e_\theta(z_t, t, c) - e_\theta(z_t, t)), \qquad (1)$$

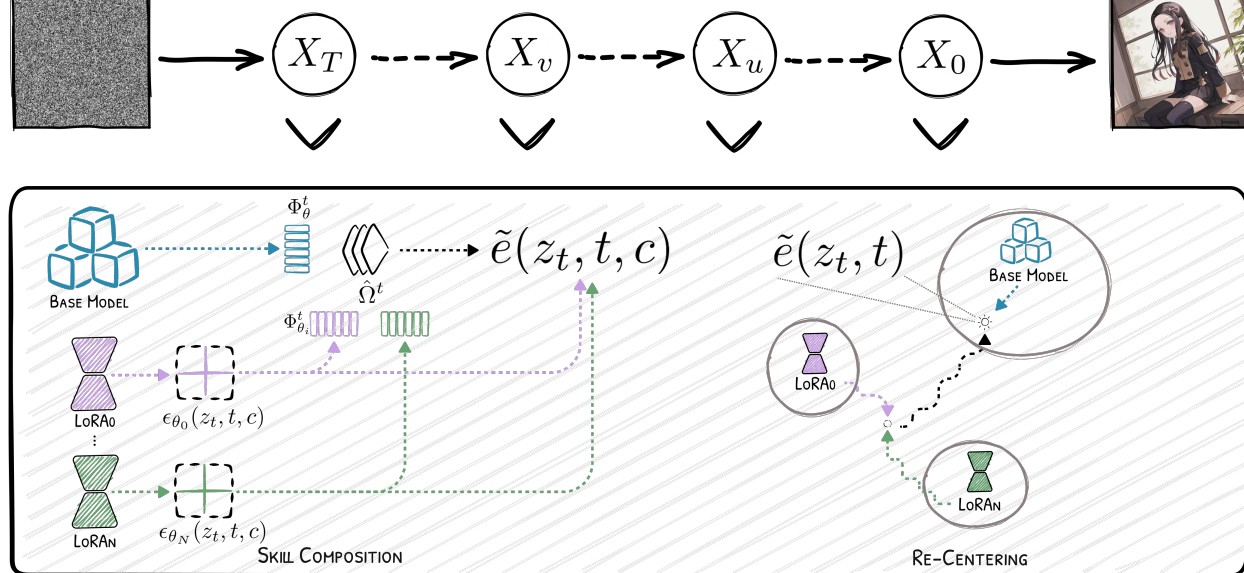

Figure 3: Overview of LORATORIO. Skill Composition: At each denoising timestep $t$, the conditional score from the $i^{\text{th}}$ LoRA, $e_{\theta_i}(z_t, t, c)$, is partitioned into $P$ spatial patches. Each patch is flattened and compared to its corresponding patch in the base model's predicted noise using cosine similarity. The resulting similarity matrix is passed through a SoftMin function to produce a weight matrix $\Omega$, assigning higher weights to patches that diverge more from the base model. These weights are used to compute a spatially-aware weighted average across LoRA outputs. Re-Centering: To mitigate domain drift of the unconditional score by the multiple LoRAs we alter classifier-free guidance by incorporating a weighted combination of the base model's unconditional score and the aggregated LoRA score.

where $s \geq 1$ is the guidance scale and $e_\theta(z_t, t)$ denotes the unconditional prediction. This approach allows the model to maintain sample diversity while enhancing conditional fidelity without relying on external classifiers.

Low-Rank Adaptation (LoRA) (Hu et al., 2022) is a parameter-efficient fine-tuning method that enables adaptation of large models by injecting trainable low-rank matrices into existing weight layers. In the context of diffusion models, LoRA allows modification of the model's behaviour (e.g. , emphasising identity features) without altering the full parameter set, thereby reducing the risk of catastrophic forgetting. Specifically, for a weight matrix $W \in \mathbb{R}^{d \times k}$, LoRA introduces a trainable update $\Delta W = AB$, where $A \in \mathbb{R}^{d \times r}$ and $B \in \mathbb{R}^{r \times k}$, with $r \ll \min(d, k)$. This decomposition allows the model to adapt key features—such as identity attributes—by updating only a small number of parameters; however, when multiple LoRA adapters are present, a linear combination of the weights may lead to semantic conflicts and reduced image quality (Huang et al., 2024a; Zhong et al., 2024; Zou et al., 2025). To address issues related to weight manipulation techniques, Zhong et al. (2024) proposes aggregating conditional and unconditional scores using a weighted average, so that for $N$ LoRAs:

$$\tilde{e}(z_t, t, c) = \frac{1}{N} \sum_{i=0}^{N} w_i \cdot \left[ e_{\theta_i}(z_t, t) + s \cdot (e_{\theta_i}(z_t, t, c) - e_{\theta_i}(z_t, t, c)) \right] \quad (2)$$

where the weights $w$ are a scalar hyperparameter (set to 1).

## 2.1 SKILL COMPOSITION USING INTRINSIC KNOWLEDGE

We propose **LoRAtorio**, a method that activates all LoRAs at each timestep by leveraging the similarity between their noise latent representations and that of the base model. Motivated by Observation 2, we compute the cosine similarity between the output of the model after incorporating the $i^{th}$ LoRA, denoted by $e_{\theta_i}(z_t, t, c)$, and the base model's output $e_\theta(z_t, t, c)$.

Since the conditioned latent representations $e(z_t, t, c)$ retain spatial structure, we first perform channel-wise averaging to reduce the dimensionality from $H \times W \times C$ to $H \times W$. We then partition each of these 2D maps into $P$ non-overlapping patches of equal size and flatten each patch into a vector. Let $\phi(\cdot) : R^{H \times W} \to R^{P \times d}$ denote this process, mapping a $H \times W$ feature map into a set of $P$ vectors in $\mathbb{R}^d$, where $d^2$ is the number of pixels per patch. We denote the resulting tokenised outputs as:

$$\Phi_\theta^t = \phi(e_\theta(z_t, t, c)), \quad \Phi_{\theta_i}^t = \phi(e_{\theta_i}(z_t, t, c)) \tag{3}$$

with $\Phi_\theta^t, \Phi_{\theta_i}^t \in \mathbb{R}^{P \times d}$. For each LoRA $i$, we compute the cosine similarity between corresponding patch vectors of $\Phi_\theta^t$ and $\Phi_{\theta_i}^t$, resulting in a weight matrix $\Omega^t = \left\langle \Phi_\theta^t, \Phi_{\theta_i}^t \right\rangle_{\cos} \in \mathbb{R}^{N \times P}$ where $N$ are the number of LoRAs. We then apply a SoftMin operation along the $N$ dimension:

$$\hat{\Omega}^t = \mathrm{softmin}_\tau(\Omega^t), \quad \text{where} \quad \mathrm{softmin}_\tau(x) = \frac{\exp\left(-x_i/\tau\right)}{\sum_{j=1}^N \exp\left(-x_j/\tau\right)} \tag{4}$$

and $\tau > 0$ is the temperature parameter controlling the softness of the SoftMin. This makes the weighting interpretable as a soft attention mechanism, where LoRAs that diverge more from the base model are given higher influence in regions where they are more confident. We upscale $\hat{\Omega}^t \in \mathbb{R}^{N \times (H/d \cdot W/d)}$ to match the spatial resolution of the original feature map using a Kronecker product:

$$\hat{\Omega}^{t,\mathrm{up}} = \hat{\Omega}^t \otimes \mathbf{1}_{d \times d} \tag{5}$$

where $\mathbf{1}_{d \times d}$ is a matrix of ones. This operation effectively repeats each similarity value over a $d \times d$ block. The upscaled similarity maps are then used to modulate the expert outputs during denoising. The final conditional estimate is computed as a weighted combination of expert predictions:

$$\tilde{e}(z_t, t, c) = \sum_{i=1}^N \hat{\Omega}_i^{t,\mathrm{up}} e_{\theta_i}(z_t, t, c) \tag{6}$$

We interpret cosine similarity in the noise prediction in the latent space as a proxy for LoRA confidence or relevance: patches where LoRA strongly deviates from the base model are assumed to reflect greater domain-specific influence. This is grounded in Observation 2 that LoRA outputs remain close to the base model when operating out-of-distribution. A theoretical motivation for similarity-based weighting is included in Appendix A

## 2.2 RE-CENTERING GUIDANCE

To address the bias of the unconditioned noise output of the model in Observation 1, we propose incorporating the output of the base model. When a set of LoRA adapters $\theta_i$ is integrated into a diffusion model, each adapter implicitly encodes the data distribution $p_{\mathrm{LoRA}i}(x)$ used during its training. As a result, the unconditional noise output $e_{\theta_i}(z_t, t)$ of the LoRA-integrated model diverges from the base model's unconditional distribution $e_\theta(z_t, t)$, which approximates the score of the base data distribution $p(x)$, empirically shown in Figure 1. Given that CFG relies on extrapolation between unconditional and conditional noise Equation (1), this mismatch introduces a "drift" in the implied guidance trajectory. Specifically, the guidance term $e_{\theta_i}(z_t, t, c) - e_{\theta_i}(z_t, t)$ is no longer a faithful estimator of the score $\nabla_x \log p(x|c) - \nabla_x \log p(x)$, but is skewed by the semantics and biases of $p_{\mathrm{LoRA}_i}(x)$. When multiple LoRAs are activated simultaneously, the unconditional outputs can conflict due to semantic incompatibility between the LoRA-specific data distributions, leading to lower subject fidelity under standard CFG.

To mitigate this drift, we propose "re-centering" the guidance computation by incorporating the unconditional base model output. Specifically, we use the average of the base model and LoRA-weighted unconditional outputs in CFG so that the final collective guidance $\hat{e}(z_t, t, c)$ is then calculated as follows:

$$\tilde{e}(z_t, t) = \lambda \sum_{i=0}^N \hat{\Omega}_i^{t,\mathrm{up}} e_{\theta_i}(z_t, t) + (1 - \lambda) e_\theta(z_t, t)$$

$$\hat{e}(z_t, t, c) = \tilde{e}(z_t, t) + s\left[\tilde{e}(z_t, t, c) - \tilde{e}(z_t, t)\right] \tag{7}$$

we set re-centering scale hyperparameter $\lambda = 0.5$, for simplicity, in all experiments. A visual representation of the re-centering method can be seen in Figure 4.

### 2.3 Dynamic module selection

The MultiLoRA composition task is defined under the assumption that only a known subset of LoRA adapters—those relevant to the current generation task—are loaded. This restricts flexibility, as it requires prior knowledge of which LoRAs are needed, and contradicts the goal of a truly inference-time, modular composition system. We propose expanding the task to a dynamic selection setting, where all available LoRA adapters are loaded into the model that dynamically selects which ones to activate based on the input. To address the dynamic setting, we propose using only the top-$k$ most distant LoRAs at each timestep $t$. We first perform a hard masking step by selecting the top-$k$ most relevant LoRA experts using a similarity-based gating metric $\Omega^t$. Specifically, we compute:

$$\mathcal{I}_k = \text{TopK}(1 - \Omega^t, k)$$

$$\tilde{\Omega}_i^t = \begin{cases} \Omega_i^t & \text{if } i \in \mathcal{I}_k \\ \infty & \text{otherwise} \end{cases} \tag{8}$$

$$\hat{\Omega}_i^t = \text{softmin}_\tau(\tilde{\Omega}^t)$$

The $\hat{\Omega}_i^t$ is the upscaled and reshaped as described in Section 2.1, so that it can be used in the subsequent weighted average and re-centering steps.

Figure 4: Visualisation of the effect of re-centering guidance on the unconditional noise score. Re-centering ensures the $p(x|c)$ is not over-emphasising implausible or under-trained regions of the collective data distribution after CFG. When the scores are similar, the transformation is not significant, but when there is a large adjustment, the difference is in the direction towards more probable samples.

## 3 Experimental Results

### 3.1 Implementation Details

For our experiments, we follow the setup of Zhong et al. (2024), using *stable-diffusion-v1.5* (Rombach et al., 2022) as the backbone for all *ComposLoRA* tests. We use the "Realistic_Vision_V5.1" and "Counterfeit-V2.5" checkpoints for realistic and anime-style images, respectively. For experiments with a Flux base model (Labs, 2024), we use the "black-forest-labs/FLUX.1-dev" checkpoint. For the realistic subset, we use 100 denoising steps, a guidance scale $s = 7$, and image size $1024 \times 768$; for the anime subset, we use 200 steps, $s = 10$, and $512 \times 512$ resolution. DPM-Solver++ (Lu et al., 2022) is used as the sampler, with all LoRAs scaled by a weight of 0.8. We empirically set an adaptive temperature $\tau = 1/((T - t) * 10)$. For all experiments, we set the size of each patch to $2 \times 2$. Since our method operates at inference time, all experiments are run on a single RTX A6000 GPU. Results are averaged over three runs.

### 3.2 CLIPScore

We employ CLIPScore (Hessel et al., 2021) to evaluate how well the generated images match the text prompt, shown in Table 1. Even though CLIPScore does not evaluate compositional quality, acting more as a bag of words (Zhong et al., 2024), it is still an important indicator of text-to-image fidelity. LoRAtorio outperforms or performs comparably to all previous methods across all $N$. Specifically, we see that with the exception of $N = 2$, where our method achieves comparable scores to previous work, LoRAtorio outperforms previous SoTA. In addition, our method does not deteriorate as $N$ increases, peaking at $N = 4$ where it outperforms previous SoTA by over 1%, proving robustness as more skills are added. A breakdown by subset can be seen in Appendix B, and an ablation of our method's components is presented in Appendix B.1.

Table 1: CLIPScore of LoRAtorio against previous composition methods on *ComposLoRA*.

| Model | $N = 2$ | $N = 3$ | $N = 4$ | $N = 5$ | Avg. |
|---|---|---|---|---|---|
| Naïve (Rombach et al., 2022) | 35.014 | 34.927 | 34.384 | 33.809 | 34.534 |
| Merge (Hugging Face, 2024) | 33.726 | 34.139 | 33.399 | 32.364 | 33.407 |
| Switch (Zhong et al., 2024) | 35.394 | 35.107 | 34.478 | 33.475 | 34.614 |
| Composite (Zhong et al., 2024) | 35.073 | 34.082 | 34.802 | 32.582 | 34.135 |
| LoraHub (Huang et al., 2024a) | 35.681 | 35.127 | 34.970 | 33.485 | 34.816 |
| Switch-A (Zou et al., 2025) | 35.451 | 35.383 | 34.877 | 33.366 | 34.769 |
| CMLoRA (Zou et al., 2025) | 35.422 | 35.215 | 35.208 | 34.341 | 35.047 |
| MultLFG (Roy et al., 2025) | **36.570** | 36.125 | 36.180 | 35.920 | 36.199 |
| LoRAtorio | 35.236 | **36.426** | **37.136** | **36.626** | **36.356** |

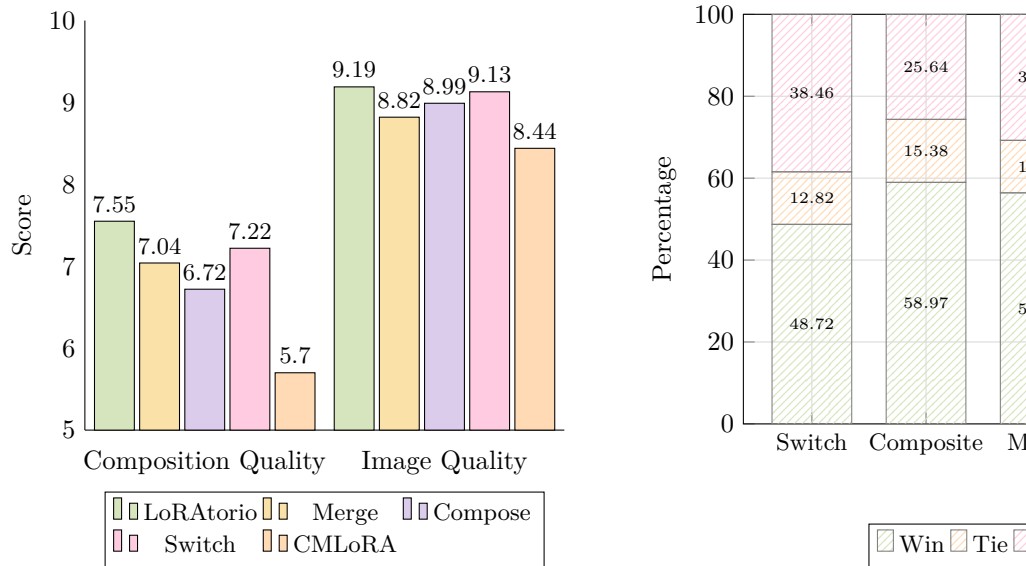

(a) Composition and Image Quality of LoRAtorio against previous SoTA.

(b) Overall win rate of LoRAtorio against previous skill composition works

Figure 5: GPT4V Evaluation on *ComposLoRA*

### 3.3 GPT4V EVALUATION

To assess the compositional and aesthetic qualities of our method, we employ a GPT4V-based evaluation as outlined in *ComposLoRA* testbed (Zhong et al., 2024), against previous SoTA where their code or images for evaluation have been made publicly available. The evaluation involves scoring LoRAtorio against Switch, Composite, Merge and CMLoRA across two dimensions, "Composition Quality" and "Image Quality". Scores and Win Rates can be seen in Figure 5a and Figure 5b, respectively. LoRAtorio outperforms previous works both in terms of average scores and win rate, i.e. pairwise comparison, closely followed by Switch. Additional results can be seen in Appendix B.

### 3.4 HUMAN EVALUATION

Further to the GPT4 evaluation, we employ human experts to assess LoRAtorio qualitatively against previous works, as described by Zou et al. (2025) across four criteria: Element Integration, Spatial Consistency, and Semantic Accuracy. The results shown in Table 2 corroborate the GPT4v evaluation, with LoRAtorio outperforming all previous works closely followed by Switch. Details of the interface and definitions for the human evaluation can be seen in Appendix G.

Table 2: Human Evaluation of our Method against previous SoTA along four qualitative axis.

|            | Element Integration | Spatial Consistency | Semantic Accuracy | Aesthetic Quality |
|------------|---------------------|---------------------|-------------------|-------------------|
| LoRAtorio  | **7.64**            | **7.58**            | **7.33**          | **6.83**          |
| CMLora     | 5.63                | 5.58                | 6.08              | 5.25              |
| Compose    | 6.46                | 6.71                | 6.71              | 6.46              |
| Switch     | 7.57                | 7.50                | 6.88              | 6.71              |
| Merge      | 6.83                | 6.71                | 6.58              | 6.08              |

### 3.5 Dynamic Module Selection

As all the LoRAs are added for the dynamic module setting, we observe that the output images of LoRA Merge become non-sensical, which is reflected both in the CLIPScore of Table 3 and qualitative output in Appendix B. Even with functionally sparse weights and limited activation, when the conditions are out of distribution, the denoising process is affected by the presence of multiple LoRAs, highlighting the need for a method that reliably selects only a relevant subset at each step. LoRAtorio maintains high CLIPScore on the dynamic setting, with minimal influence from unrelated LoRAs as can be visually verified in Appendix B.

Table 3: CLIPScore of LoRAtorio against previous SoTA on *ComposLoRA* in a dynamic module selection setting, where $N$ is the number of LoRA experts needed.

|            | $N = 2$  | $N = 3$  | $N = 4$  | $N = 5$  | Avg.     |
|------------|----------|----------|----------|----------|----------|
| LoRAtorio  | **34.593** | **35.563** | **36.480** | **37.028** | **35.916** |
| Naïve      | 35.014   | 34.927   | 34.384   | 33.809   | 34.534   |
| Merge      | 27.167   | 27.151   | 27.023   | 27.272   | 27.153   |

### 3.6 Flux

As our method is model agnostic and can be implemented in any latent diffusion method, we present results with a Rectified Flow (Liu et al., 2023) base model. As Flux 1.D is using a transformer-based architecture to produce $e_\theta$, we omit the tokenisation and re-centering step. As seen by the CLIPScore in Table 4, LoRAtorio significantly outperforms the baselines and shows consistent improvement as $N$ increases, attributed to longer text conditions. This trend is consistent in both the static and dynamic module settings, corroborating the results of the SD1.5 experiments. Details on the prompts and LoRAs used for experiments using Flux architecture can be seen in Appendix E.

Table 4: CLIPScore of LoRAtorio against selected composition methods, using Flux architecture.

(a) Static Modules

| Model     | $N = 2$ | $N = 3$ | $N = 4$ | $N = 5$ | Avg.   |
|-----------|---------|---------|---------|---------|--------|
| Naïve     | 33.125  | 34.999  | 37.048  | 38.568  | 35.935 |
| Merge     | 33.733  | 35.134  | 35.830  | 36.590  | 35.322 |
| LoRAtorio | **33.992** | **36.033** | **37.781** | **39.368** | **36.794** |

(b) Dynamic Module Selection

| Model     | $N = 2$ | $N = 3$ | $N = 4$ | $N = 5$ | Avg.   |
|-----------|---------|---------|---------|---------|--------|
| Naïve     | 33.125  | 34.999  | 37.048  | 38.568  | 35.935 |
| Merge     | 25.850  | 27.858  | 29.726  | 30.997  | 28.608 |
| LoRAtorio | **33.284** | **35.753** | **37.910** | **38.861** | **36.452** |

## 4 Related Work

### 4.1 Text-to-image Generation

Composable image generation is a central challenge in personalised content creation, where the goal is to synthesise images that faithfully integrate multiple user-specified concepts. Early approaches focused on layout- or scene-graph-based conditioning to improve compositionality (Johnson et al., 2018; Song et al., 2021; Gafni et al., 2022). More recent work has shifted toward modifying the generative process of diffusion models to better align with structured or multi-concept prompts (Feng et al., 2023; Huang et al., 2023; Kumari et al., 2023; Lin et al., 2023; Ouyang et al., 2025). These methods often rely on prompt engineering

or architectural changes to enforce compositional constraints and struggle with precise integration of user-defined elements such as rare characters, styles, or objects. Some methods address this by composing multiple independently trained modules (Du et al., 2020; Liu et al., 2021; Li et al., 2023; Simsar et al., 2025), but they often require extensive fine-tuning and do not scale well with the number of concepts. Our work builds on this line by proposing a train-free, instance-level composition framework that leverages LoRA adapters to enable fine-grained, spatially-aware integration of multiple concepts.

## 4.2 LoRa-Based SKill Composition

Low-Rank Adaptation (LoRA) has emerged as a lightweight and effective method for fine-tuning large models, including diffusion models, for personalisation tasks (Ruiz et al., 2023; Sohn et al., 2023). Recent research has explored various strategies for composing LoRA adapters to support multi-concept generation.

LoRAHub (Huang et al., 2024a) and ZipLoRA (Shah et al., 2024) use few-shot demonstrations to learn a coefficient matrix that linearly combines the weights of multiple LoRAs. This enables the creation of a new LoRA that approximates the behaviour of the original set, while reducing memory and compute overhead. Similarly, Zhu et al. (2024) propose a trainable mixture-of-experts framework, where each LoRA acts as an expert and a gating network learns to combine their outputs. Hypernetwork-based approaches (Shenaj et al., 2024; Ruiz et al., 2024) introduce a hypernetwork that generates LoRA weights conditioned on the target composition. These methods often require additional training data or supervision, and may not generalise well to open-vocabulary or zero-shot settings. LoRA Merge (Hugging Face, 2024) performs weight-level arithmetic operations to combine multiple LoRAs. CLoRA (Meral et al., 2024) improves upon attention map manipulation by comparing the attention maps to sub-sets of the text condition. Other approaches, such as LoRA Switch and LoRA Composite (Zhong et al., 2024), avoid merging weights and instead manipulate the inference process by alternating or aggregating LoRA outputs at each denoising step. MultLFG (Roy et al., 2025) employs frequency-domain guidance to fuse multiple LoRAs; however, this approach necessitates decoding at each step, making it inherently slow and computationally expensive. In addition, by decoding the images, the approach is in practice using RGB-based frequency rather than intrinsic knowledge of the network. Furthermore, while the end output of the diffusion process is indeed an image, the prediction itself is noise, therefore our work takes a more intuitive approach by exploring latent space noise predictions instead of RGB frequency. Similarly, Zou et al. (2025) expands LoRA-Composite with frequency-based scheduling and introduces a caching mechanism. While effective, these methods often suffer from instability and semantic conflicts as the number of LoRAs increases. Additionally, they do not explicitly account for the interaction between LoRA outputs and the base model, do not account for domain shift from LoRA fine-tuning and are limited to text conditions.

Our method draws inspiration from spatial composition techniques such as CutMix (Yun et al., 2019) and token-level fusion (Wang et al., 2024), but applies these principles in the latent space for image generation in a train-free setting. While LoRAtorio resembles mixture-of-experts (Jacobs et al., 1991) in spirit, it differs in three key ways: (1) it uses intrinsic cosine similarity between LoRA and base model latents for gating, rather than learned or supervised routing; (2) its patchwise weighting operates in the semantic latent space of the diffusion model, rather than in image or feature space; and (3) it requires no fine-tuning, supervision, or additional modules, enabling zero-shot, inference-time composition of arbitrary LoRA adapters.

## 5 Conclusion

In this paper, we present a novel, train-free approach to multi-LoRA composition through the introduction of LoRAtorio, a method grounded in intrinsic model behaviour. Motivated by empirical observations of domain drift and latent-space divergence, our method leverages spatially-aware cosine similarity to dynamically weight LoRA contributions at the patch level. We further propose a modification to classifier-free guidance that incorporates the base model's unconditional signal, improving robustness in out-of-domain scenarios. Extending beyond static composition, we formulate the task as one of dynamic module selection, enabling inference-time adaptability in settings where irrelevant skills are loaded to the base model. Our approach achieves state-of-the-art performance and generalises to Rectified Flow models.

## 6 ETHICS STATEMENT

This work examines the capabilities of generative AI models, including those enhanced with community-provided LoRAs. While generative tools offer valuable opportunities for creative and technical innovation, they also carry significant risks, including misuse for deceptive content, reinforcement of harmful biases, and uncertainty around authorship and licensing.

We do not condone the misuse of generative models, including for misinformation, harassment, or any activity that infringes on the rights of others. This work is not licensed or intended for commercial or for-profit use. We encourage future users and researchers to carefully consider the ethical and legal implications of models or data.

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

Table 5: ClipScores for *ComposLoRA* on anime and reality subsets.

(a) Anime – Static Modules

| Model | $N=2$ | $N=3$ | $N=4$ | $N=5$ | Avg. |
|---|---|---|---|---|---|
| Merge | 35.136 | 35.421 | 34.164 | 32.636 | 34.339 |
| Switch | 35.285 | 35.482 | 34.532 | 34.148 | 34.861 |
| Composite | 34.343 | 34.378 | 34.161 | 32.936 | 33.955 |
| LoraHub | 35.316 | 35.525 | 34.476 | 33.885 | 34.801 |
| Switch-A | 35.705 | 35.912 | 35.661 | 34.479 | 35.439 |
| CMLoRA | 35.556 | 35.555 | 35.791 | 35.691 | 35.648 |
| MultLFG | **36.720** | 36.130 | 36.450 | 36.220 | 36.380 |
| LoRAtorio | 36.156 | **36.930** | **36.864** | **36.162** | **36.528** |

(b) Reality – Static Modules

| Model | $N=2$ | $N=3$ | $N=4$ | $N=5$ | Avg. |
|---|---|---|---|---|---|
| Merge | 32.316 | 32.857 | 32.633 | 32.091 | 32.474 |
| Switch | 35.502 | 34.731 | 34.424 | 32.801 | 34.365 |
| Composite | 35.804 | 33.786 | 35.443 | 32.228 | 34.315 |
| LoraHub | 36.045 | 34.729 | 35.463 | 33.084 | 35.412 |
| Switch-A | 35.196 | 34.854 | 34.694 | 32.252 | 34.249 |
| CMLoRA | 35.559 | 35.842 | 34.501 | 33.588 | 34.873 |
| MultLFG | **36.420** | 36.120 | 35.910 | 35.620 | 36.018 |
| LoRAtorio | 34.316 | **35.922** | **37.408** | **37.090** | **36.184** |

(c) Anime – Dynamic Modules

| | $N=2$ | $N=3$ | $N=4$ | $N=5$ | Avg. |
|---|---|---|---|---|---|
| LoRAtorio | **35.328** | **35.931** | **36.332** | **36.184** | **35.944** |
| Naïve | 35.014 | 34.927 | 34.384 | 33.809 | 34.534 |
| Merge | 30.953 | 30.767 | 30.352 | 30.488 | 30.640 |

(d) Reality – Dynamic Modules

| | $N=2$ | $N=3$ | $N=4$ | $N=5$ | Avg. |
|---|---|---|---|---|---|
| LoRAtorio | 33.858 | **35.194** | **36.627** | **37.871** | **35.888** |
| Naïve | **35.014** | 34.927 | 34.384 | 33.809 | 34.534 |
| Merge | 23.381 | 23.534 | 23.693 | 24.055 | 23.666 |

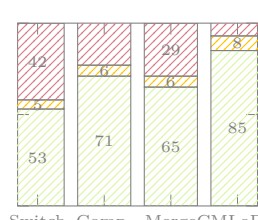
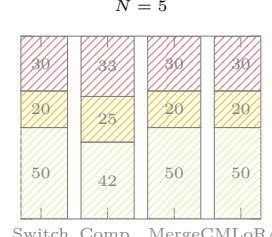

Figure 6: Win/Tie/Loss for LoRAtorio compared to previous SoTA across different number of LoRAs ($N$).

## A  THEORETICAL MOTIVATION FOR SIMILARITY-BASED WEIGHTING.

LoRA introduces a low-rank update to a weight matrix $W \in \mathbb{R}^{d \times k}$ in the form $\Delta W = AB$, where $A \in \mathbb{R}^{d \times r}$, $B \in \mathbb{R}^{r \times k}$, and $r \ll \min(d, k)$ (Hu et al., 2022). This constrains the update to lie in a low-dimensional subspace of the weight space, limiting the directions in which the model can adapt. Such low-rank adaptation has been shown to improve parameter efficiency and mitigate catastrophic forgetting (Biderman et al., 2024).

More precisely, the LoRA update acts on inputs $x \in \mathbb{R}^k$ by first projecting via $A$, $Ax \in \mathbb{R}^r$, then mapping back to output space via $B$. The effective input subspace to which the adapter responds is the row space of $A$, i.e. , inputs $x$ for which $Ax \neq 0$. For inputs $x'$ approximately orthogonal to this subspace, $Ax' \approx 0$ and thus

$$\Delta W x' = (Ax')B \approx 0, \tag{9}$$

implying

$$W x' + \Delta W x' \approx W x'. \tag{10}$$

Hence, the LoRA adapter has a negligible effect on inputs lying outside its learned subspace, which often correspond to out-of-distribution (OOD) inputs, and subsequent non-linearities in deep learning models further mitigate the effect of LoRAs. Consequently, the latent outputs of the LoRA-augmented model and the base model are similar for OOD inputs. This motivates using the cosine similarity between their latent outputs as a proxy for the adapter's confidence or relevance: high similarity indicates that the adapter is inactive or uncertain (OOD), whereas lower similarity suggests in-distribution behaviour where the adapter actively modifies the model output. This observation underpins our use of cosine similarity in LoRAtorio.

## B  EXPERIMENTAL RESULTS

Further to the main experimental results in Section 3, we show ClipScores for the subsets of *ComposLoRA* in Table 5 for anime and reality subsets in both static and dynamic module settings. LoRAtorio maintains

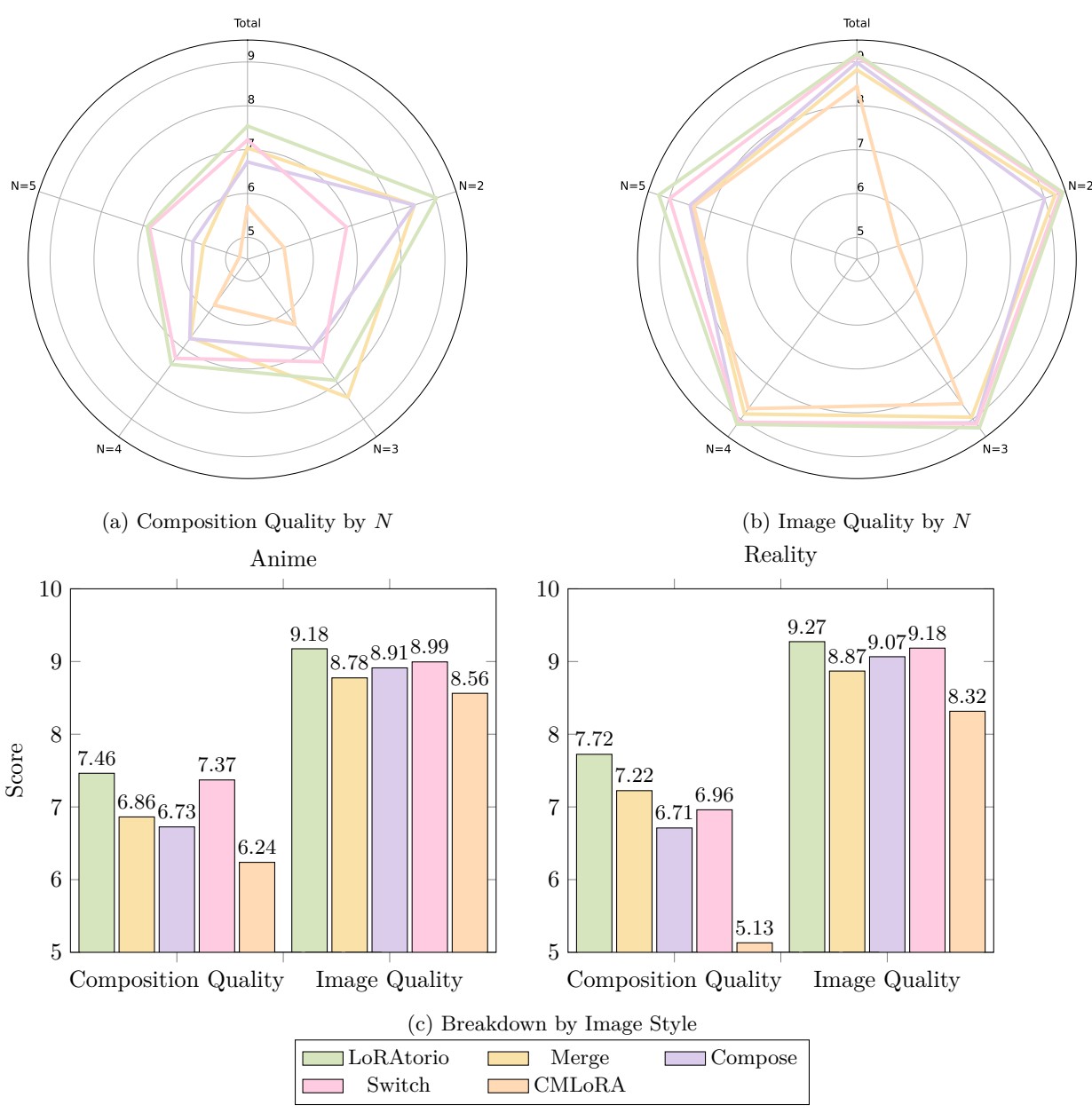

(a) Composition Quality by $N$

(b) Image Quality by $N$

(c) Breakdown by Image Style

Figure 7: Composition and Image Quality scores of LoRAtorio against previous SoTA on *ComposeLoRA*, by number of LoRA's included and image style.

strong performance in both subsets, having the highest weighted average score compared to previous SoTA. This is consistent for both static and dynamic module selection. We observe similar trends in performance within each subset for the number of LoRAs included, with LoRAtorio clearly outperforming other works on average. As expected, we also see weight merge collapsing in the dynamic module setting.

Further to the ClipScores, we present the GPT4v evaluation results by number of LoRAs included and by sub-set in 7. LoRAtorio maintains robust performance in all scenarios, showing strong composition and image quality. Finally, we include the win rate of our method by number of LoRAs included, showing SoTA performance, particularly as $N$ increases in Figure 6.

## B.1 ABLATION STUDY

To show the effect of LoRAtorio's components, we perform an ablation study as shown in Table 6, using CLIPScore as an evaluation metric. Note that CLIPScore is unable to capture the composition of aesthetic quality, so the final set of hyperparameters is selected as a combination of CLIPScore and empirically through visual inspection of output. Specifically, we compute the CLIPScore of generated images using the distance of the entire image instead of individual patches, with a constant $\tau = 1$ and without our re-centering method. The localised activation of LoRAs through the tokenisation of the latent space has the greatest impact in terms of CLIPScore , which is somewhat expected as more elements can be integrated and thus aligned in clip space. We also compare the effect of patch size on the performance of LoRAtorio and see that a more fine-grained composition results in higher CLIPScore , although the performance is relatively robust.

Table 6: Ablation study of our method on *ComposLoRA* Anime subset, using CLIPScore .

(a) LoRAtorio Components

|  | $N = 2$ | $N = 3$ | Avg. |
|---|---|---|---|
| LoRAtorio | 36.156 | **36.930** | **36.543** |
| w/o $\phi(\cdot)$ | 34.306 | 33.967 | 34.137 |
| w/o $\tau$ | 35.948 | 36.690 | 36.319 |
| w/o Re-centering | **36.477** | 36.586 | 36.532 |

(b) Patch Size

|  | $N = 2$ | $N = 3$ | Avg. |
|---|---|---|---|
| $2 \times 2$ | **36.156** | **36.930** | **36.543** |
| $4 \times 4$ | 36.025 | 36.475 | 36.250 |
| $8 \times 8$ | 35.852 | 36.423 | 36.138 |
| $16 \times 16$ | 35.744 | 36.003 | 35.874 |

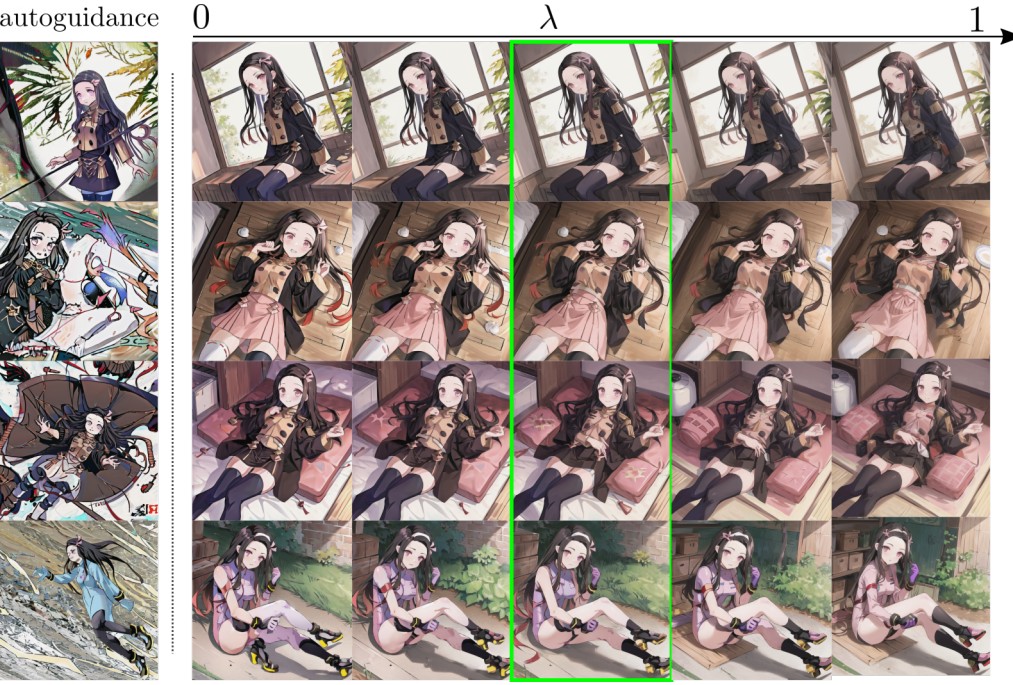

Figure 8: Qualitative comparison of LoRAtorio's re-centering guidance across different values of $\lambda$, evaluated against auto-guidance (Karras et al., 2024). The figure illustrates the impact of varying $\lambda$ on image coherence, identity preservation, and visual quality.[1]

In addition, we conduct a qualitative comparison of LoRAtorio's re-centering guidance with auto-guidance (Karras et al., 2024), and evaluate performance across different values of the weighting parameter $\lambda$, shown on Figure 8, as CLIPScore alone does not capture identity preservation, compositionally or other qualitative elements. More specifically, we show in Figure 8 that the CLIPScore is consistent for all

---

[1]CLIPScore for the images in each row is identical, including autoguidance samples highlighting the necessity of visual inspection and qualitative evaluation.
Row 1:32.552, Row 2: 31.810, Row 3: 31.763, Row 4: 32.653

selected images, thus reiterating that it should be used a metric of generic object inclusion not instance fidelity or qualitative score. Since the base model is essentially "a bad version" of the LoRA-augmented model, auto-guidance serves as a natural baseline for assessing our re-centering approach. Notably, we find that combining LoRAtorio with auto-guidance fails to produce coherent images. We hypothesise this is due to a difference in data distribution, a prerequisite for auto-guidance. Similarly, when $\lambda = 0$ – where the unconditioned score corresponds to that of the base model – we observe strong identity preservation, but the resulting images exhibit excessive saturation and appear unnatural. Conversely, setting $\lambda = 1$ results in some loss of identity and a blending of concepts. To balance these effects, we empirically select $\lambda = 0.5$ for all experiments, for simplicity.

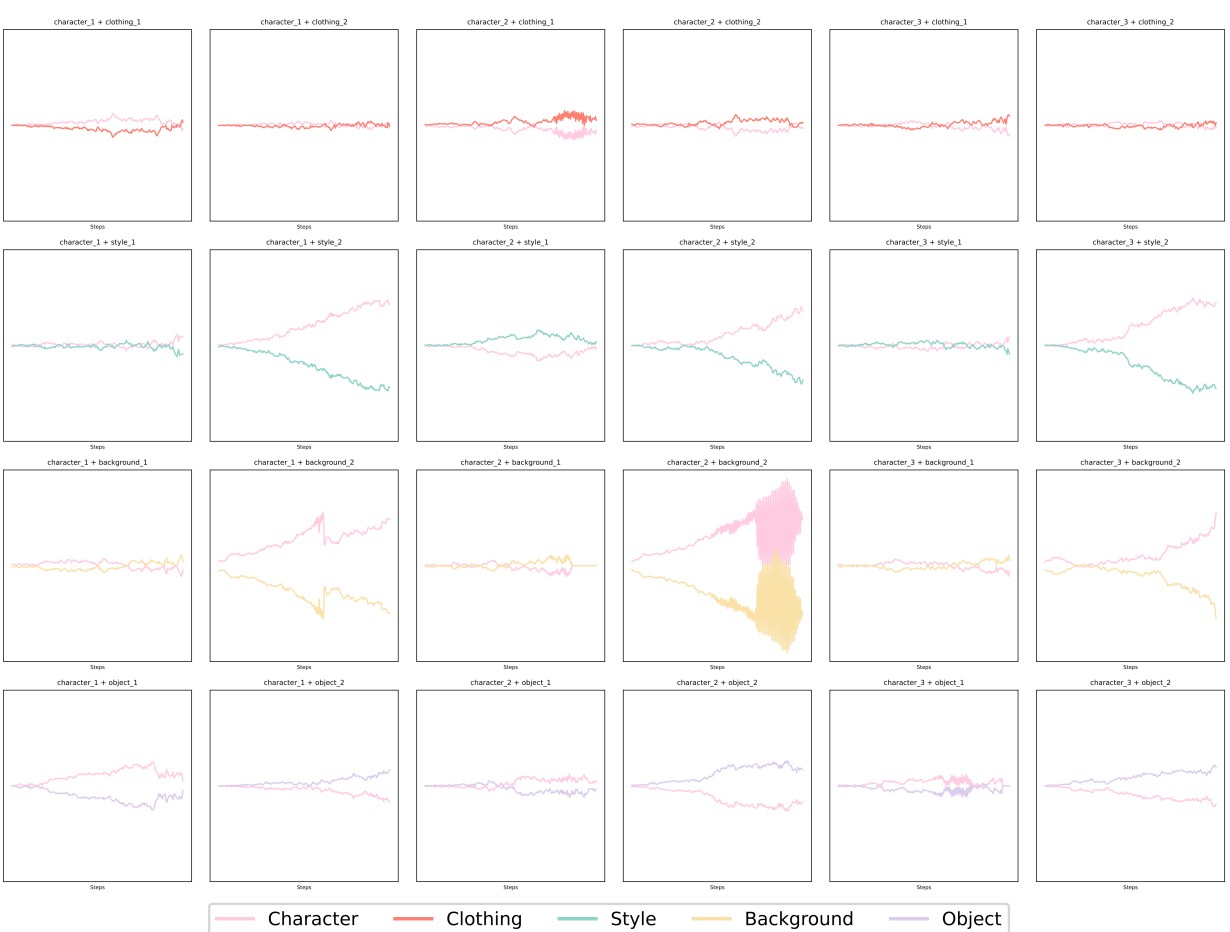

Figure 9: Average values of $\Omega^t$ over denoising process, for the entire image in the *ComposLoRA* testbed for $N = 2$.

## C  TEMPORAL ANALYSIS OF SIMILARITY-BASED WEIGHTING

Our similarity-based weighting mechanism is designed as a proxy of the relative confidence of each LoRA adapter with respect to the base model. Empirically, we observe that the cosine similarity between LoRA-augmented outputs and the base model varies non-uniformly across denoising steps, depending on the LoRA employed.

This temporal asymmetry aligns with prior findings in diffusion literature (Si et al., 2024; Zhong et al., 2024; Zou et al., 2025), which show that different semantic attributes emerge at different stages of the denoising process. However, we observe that the variation within LoRAs of the same type (e.g. clothing or style) is too vast for universal and concrete conclusions on the order of activations. We believe this to be due to different LoRA training and configuration, further explored in Appendix D

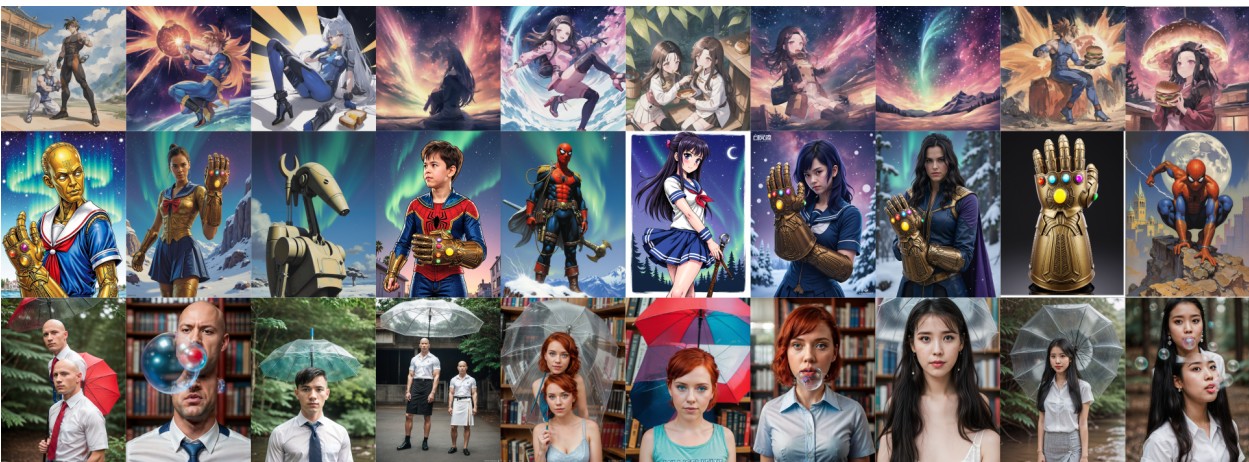

Figure 10: $\Omega^t$ map for character overlayed over the final generated images across timesteps, in Character + Style generation. As style has more global effect, we see clearly the effect of the character LoRA on the predicted noise through the $\Omega^t$ heatmap (high effect is red, no effect is transparent).

To validate this behaviour, we analyse the evolution of the similarity matrix $\Omega^t$ over time, aggregated across the latent representation. As shown in Figure 9, the pattern is not very consistent for any of the element groups. As such, methods relying on guiding based on the type of LoRA used ignore this intrinsic proxy for confidence completely. However, because different elements vary in spatial extent, a naïve global aggregation would disproportionately favour larger elements – particularly in early steps that affect the trajectory of the denoising process (Zhong et al., 2024). This motivates our use of the spatial tokenisation function $\phi$, which enables fine-grained, patch-level weighting and ensures that larger background or clothing regions do not overshadow smaller but semantically important regions (e.g. , smaller objects). A visualisation of the patchwise similarity over the diffusion process can be seen in Figure 10.

# D    LIMITATIONS AND ERROR ANALYSIS

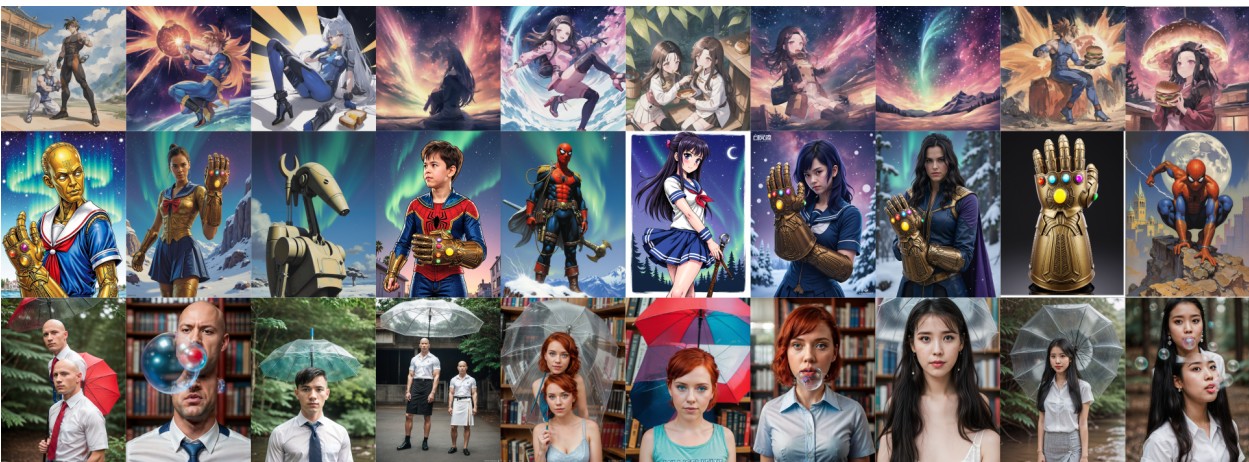

Figure 11: Examples of failure cases of LoRAtorio on *ComposLoRA* anime (top), Flux (mid) and *ComposLoRA* reality (bottom) test beds.

One key limitation of our method is the computational cost. While the intrinsic nature of LoRAtorio allows for a better understanding of the generation process and shows competitive results, the computational cost increases linearly with every additional LoRA. This limitation is identified by previous works manipulating the latent space instead of the weights (Zhong et al., 2024). This is especially true in the open-vocabulary setting where all available LoRA adapters are loaded. Potential future directions to address these limitations include exploring the subspace at an earlier stage (i.e. based on early-layer similarity, which we have not explored in the scope of this work), so that pruning or TopK can be implemented before obtaining the latent denoised output. Furthermore, model parallelisation may increase the speed of inference by estimating denoised outputs on different GPUs.

Our method assumes that LoRA adapters are trained on reasonably well-aligned and semantically coherent datasets. In practice, however, LoRA quality can vary significantly – particularly when sourced from community repositories such as CivitAI – where training data, objectives, and preprocessing pipelines are often undocumented or inconsistent. This variability can undermine the reliability of any train-free approach. Moreover, the LoRAs used during inference are heterogeneous in terms of optimal hyperparameters (e.g. ,

guidance scale, LoRA scale), and treating them uniformly may inadvertently bias the composition toward certain adapters, especially those with more aggressive or dominant activations. While we expect better performance when LoRAs are trained under similar conditions, such alignment is rarely guaranteed in user-driven settings. One potential mitigation strategy in real-life applications is to incorporate a lightweight pre-filtering step to assess LoRA quality before inclusion. Alternatively, metadata-based heuristics (e.g. , dataset size, training steps, or CFG guidance scale) could be used to cluster or filter LoRAs. Although these approaches are not explored in this work, they represent promising directions for improving robustness in real-world deployments. Finally, as all LoRAs in the *ComposLoRA* and Flux testbeds are sourced from CivitAI without access to training details, we emphasise that all results should be interpreted in light of this uncertainty.

Finally, we note that the quality of images is affected by the base model. Figure 11 shows examples of fail cases of LoRAtorio for all three base models. We observe that the Stable Diffusion backbone (top and bottom rows) exhibits more instances of additional limbs or duplicate characters compared to the Flux backbone (middle row), where most failure cases are attributed to concept confusion. As such, expanding the test bed to more backbones is essential in dissecting base model vs method limitations.

## E  FLUX TESTBED

For experiments on Flux, we select the LoRAs described in Table 7, following a selection process similar to *ComposLoRA*. All LoRAs used in the Flux experiments are publicly available through CivitAI. We select LoRAs for three characters, two clothing, two styles, two objects and one background.

Table 7: Details of LoRA adapters used in Flux experiments.

| LoRA | Category | Trigger | Source |
|---|---|---|---|
| Yennefer of Vengerberg | Character | Yennefer | Link |
| The amazing Spiderman | Character | Spider-Man, Peter Parker | Link |
| B1 Battle Droid | Character | 7-B1 droid | Link |
| Star Wars imperial officer uniform | Clothing | Wearing an imperial officer IMPOFF uniform | Link |
| Japanese school uniform - sailorfuku | Clothing | wearing a japanese school uniform sailorfuku serafuku sailor suit | Link |
| Frank Frazetta Style Oil Painting | Style | in the style of Frank Frazetta fantasy oil painting | Link |
| Engraving Style | Style | in engraving style | Link |
| Infinity Goblet | Object | with a glove like Infinity Gauntlet | Link |
| Crescent Wrench | Object | with a Crescent Wrench | Link |
| Northern Lights | Background | with Northern lights style background | Link |

## F  GPT4V EVALUATION INTERFACE

For the GPT4v evaluation, we follow the method of Zhong et al. (2024). Specifically, we do pairwise comparisons of our method against previous works. The comparison is run twice for each pair, switching the order of images to account for any bias induced from ordering. The scores are then averaged. Pseudo-code of the evaluation can be seen in Figure 12. The prompt used in the evaluation can be seen in Appendix F.1.

### F.1  GPT4V PROMPT

I need assistance in comparatively evaluating two text-to-image models based on their ability to compose different elements into a single image. The elements and their key features are as follows:

    <IMAGE_1> <IMAGE_2> <PROMPT>

Please help me rate both given images on the following evaluation dimensions and criteria:

Composition quality:

- Score on a scale of 0 to 10, in 0.5 increments, where 10 is the best and 0 is the worst.
- Deduct 3 points if any element is missing or incorrectly depicted.

```python
def evaluate():
    image_n = 196 # number of images to evaluate
    gpt4v = GPT4V()
    # Load images
    image_path = "images"
    images = []
    for i in range(0, image_n + 0):
        cur_image = encode_image(join(image_path, f"{i}.png"))
        images.append(cur_image)

    # Load prompts usd to generate the images
    prompts = []
    with open("image_info.json") as f:
        image_info = json.loads(f.read())
    for i in range(len(image_info)):
        cur_prompt = "\n".join(image_info[i]["prompt"])
        prompts.append(cur_prompt)

    # Comparative evaluation
    gpt4v = GPT4V()
    gpt4v_scores = [{} for _ in range(image_n)]
    # i:      method 1
    # i + 1: method 2
    for i in tqdm(range(0, image_n, 2)):
        method1_image    = images[i]
        method2_image    = images[i + 1]

        cur_prompt = get_eval_prompt(prompts[i])

        compare_images(method1_image, method2_image, "method_1", "method_2", gpt4v
    , cur_prompt)
        compare_images(method2_image, method1_image, "method_2", "method_1", gpt4v
    , cur_prompt)
```

Figure 12: Pseudo-code of GPT4v Evaluation. For each pair of images, the comparison is run twice to account for bias in presentation order.

- Deduct 1 point for each missing or incorrect feature within an element.
- Deduct 1 point for minor inconsistencies or lack of harmony between elements.
- Additional deductions can be made for compositions that lack coherence, creativity, or realism.

Image quality:

- Score on a scale of 0 to 10, in 0.5 increments, where 10 is the best and 0 is the worst.
- Deduct 3 points for each deformity in the image (e.g., extra limbs or fingers, distorted face, incorrect proportions).
- Deduct 2 points for noticeable issues with texture, lighting, or color.
- Deduct 1 point for each minor flaw or imperfection.
- Additional deductions can be made for any issues affecting the overall aesthetic or clarity of the image.

Please format the evaluation as follows:

For Image 1:

[Explanation of evaluation]

For Image 2:

[Explanation of evaluation]

Scores:

Image 1: Composition Quality: [score]/10, Image Quality: [score]/10

Image 2: Composition Quality: [score]/10, Image Quality: [score]/10

Based on the above guidelines, help me to conduct a step-by-step comparative evaluation of the given images. The scoring should follow two principles:

1. Please evaluate critically.
2. Try not to let the two models end in a tie on both dimensions.

## G   HUMAN EVALUATION INTERFACE

For the human qualitative evaluation, we follow the framework of Zou et al. (2025) along four qualitative axes. For each combination, we provide a set of reference images and output of the anonymised methods. Samples of the instructions and survey, as shown to human experts, are presented below. We use three human experts to evaluate our method against previous SoTA.

# Qualitative comparison

The evaluation will take 20-25 minutes.
The aim of this evaluation, is to rate the images along 4 axis. The detailed criteria of the evaluation as shown below.

IMAGE EVALUATION METRICS
1) ELEMENT INTEGRATION
Score on a scale of 0 to 10, in 1.0 increments, where 10 is the best and 0 is the worst.
Description: How seamlessly different elements are combined within the image.
Criteria:

- Visual Cohesion: Assess whether elements appear as part of a unified scene rather than disjointed parts.
- Object Overlap and Interaction: Check for natural overlaps and interactions between objects, avoiding unnatural placements or intersections.

2) SPATIAL CONSISTENCY
Score on a scale of 0 to 10, in 1 increments, where 10 is the best and 0 is the worst.
Description: Uniformity in style, lighting, and perspective across all elements.
Criteria:

- Stylistic Uniformity: All elements should share a consistent artistic style (e.g., realism, cartoonish).
- Lighting and Shadows: Ensure consistent light sources and shadow directions to maintain realism.
- Perspective Alignment: Elements should adhere to a common perspective, avoiding mismatched viewpoints.

3) SEMANTIC ACCURACY
Score on a scale of 0 to 10, in 1 increments, where 10 is the best and 0 is the worst.
Description: Correct interpretation and representation of each element as described in the prompt.
Criteria:

- Object Accuracy: Objects should match their descriptions in type, attributes, and context.
- Action and Interaction: Actions or interactions between objects should be depicted correctly.

4) AESTHETIC QUALITY
Score on a scale of 0 to 10, in 1 increments, where 10 is the best and 0 is the worst.
Description: Overall visual appeal and artistic quality of the generated image.
Criteria:

- Colour Harmony: Use of colour palettes that are visually pleasing and appropriate for the scene.
- Composition Balance: Balanced arrangement of elements to create an engaging composition.
- Clarity and Sharpness: Images should be clear, with well-defined elements and no unwanted blurriness.

Character 1 Clothing 1

Reference

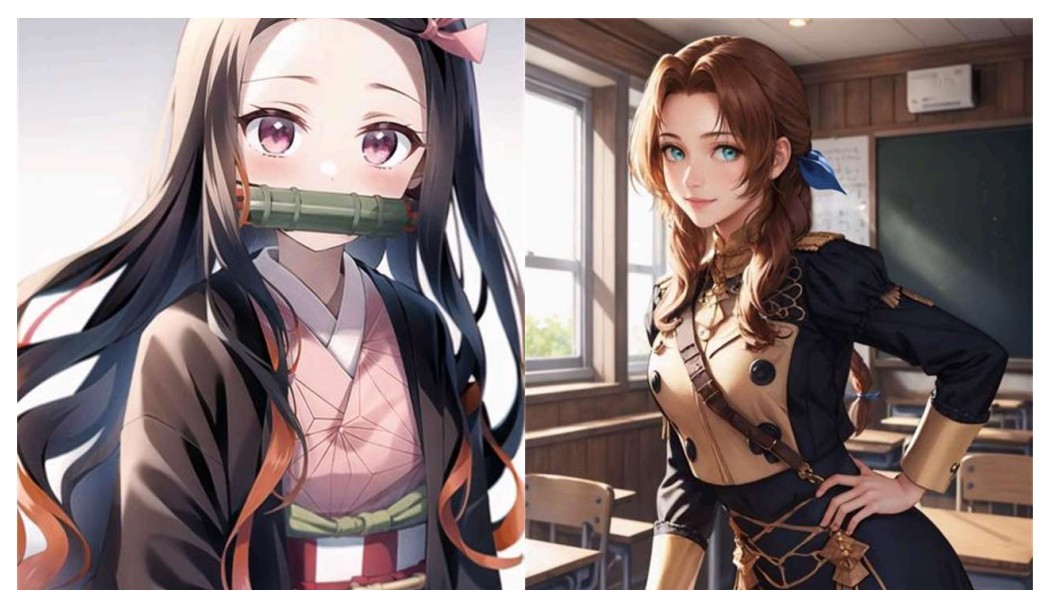

Method 1

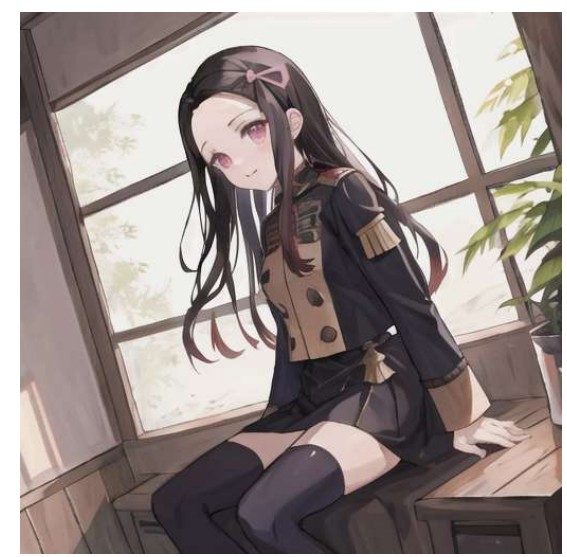

Element Integration

1    2    3    4    5    6    7    8    9    10

Spatial
Consistency

1    2    3    4    5    6    7    8    9    10

Semantic Accuracy

1    2    3    4    5    6    7    8    9    10

Aesthetic Quality

1    2    3    4    5    6    7    8    9    10

Method 2

Element Integration

     1     2     3     4     5     6     7     8     9     10

Spatial
Consistency

     1     2     3     4     5     6     7     8     9     10

Semantic Accuracy

     1     2     3     4     5     6     7     8     9     10

Aesthetic Quality

     1     2     3     4     5     6     7     8     9     10

Method 3

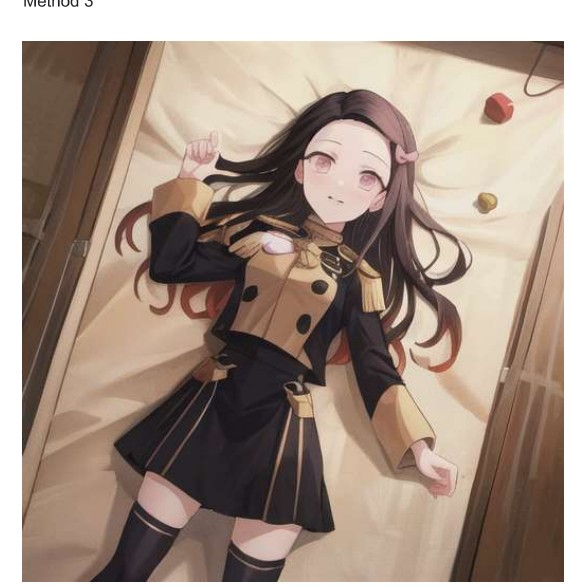

**Element Integration**

1     2     3     4     5     6     7     8     9     10

**Spatial Consistency**

1     2     3     4     5     6     7     8     9     10

**Semantic Accuracy**

1     2     3     4     5     6     7     8     9     10

**Aesthetic Quality**

1     2     3     4     5     6     7     8     9     10

**Method 4**

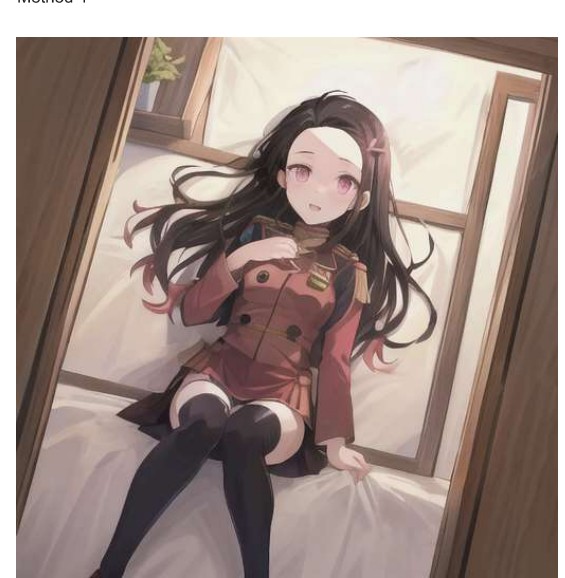

Element Integration

1   2   3   4   5   6   7   8   9   10

Spatial
Consistency

1   2   3   4   5   6   7   8   9   10

Semantic Accuracy

1   2   3   4   5   6   7   8   9   10

Aesthetic Quality

1   2   3   4   5   6   7   8   9   10

Method 5

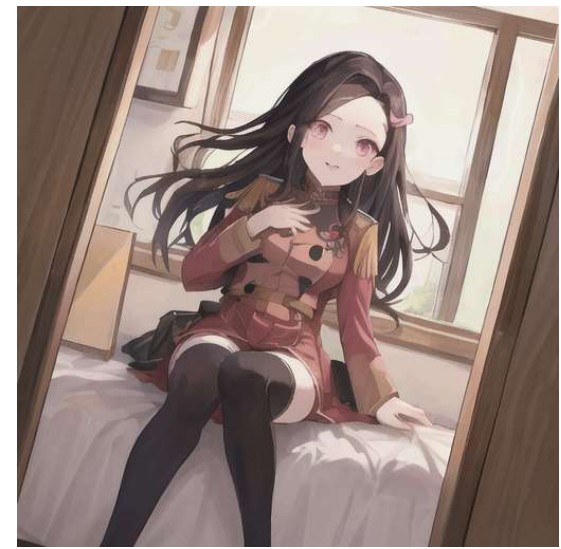

Element Integration

1    2    3    4    5    6    7    8    9    10

Spatial
Consistency

1    2    3    4    5    6    7    8    9    10

Semantic Accuracy

1    2    3    4    5    6    7    8    9    10

Aesthetic Quality

1    2    3    4    5    6    7    8    9    10

Table 8: Comparison of Multiply-Accumulate Operations (MACs) and qualitative latency estimates under different $N$ values.

(a) MACs (in GigaOps) for different methods.

|  | $N = 2$ | $N = 3$ | $N = 4$ | $N = 5$ |
|---|---|---|---|---|
| LoRAtorio | 1090.863 | 1102.721 | 1125.570 | 1132.123 |
| CMLoRA[a] | 912.350 | 1223.486 | 1358.518 | 1570.335 |
| Switch-A[a] | 734.053 | 730.914 | 739.322 | 731.811 |
| LoraHub[a] | 789.770 | 834.613 | 924.299 | 946.721 |
| Composite[a] | 1401.066 | 2169.199 | 2892.266 | 3615.333 |
| Switch[a] | 734.053 | 730.914 | 739.322 | 731.811 |
| Merge[a] | 789.770 | 834.613 | 924.299 | 946.721 |

(b) Qualitative latency estimates (seconds).

|  | $N = 2$ | $N = 3$ | $N = 4$ | $N = 5$ |
|---|---|---|---|---|
| LoRAtorio | 61 | 85 | 91 | 122 |
| Merge[b] | 20 | 21 | 22 | 24 |
| Switch[b] | 16 | 18 | 19 | 20 |
| Composite[b] | 60 | 70 | 76 | 90 |
| MultLFG[b] | 90 | 140 | 180 | 230 |

[a] As reported by Zou et al. (2025)
[b] As reported by Roy et al. (2025)

## H    COMPUTATIONAL COST ANALYSIS

To evaluate the computational efficiency of LoRAtorio, we compare the number of Multiply-Accumulate Operations (MACs) required for inference under different LoRA integration strategies and varying numbers of active adapters ($N$). Table 8a summarises the MACs for each method, highlighting the scalability and cost implications.

While LoRAtorio demonstrates competitive performance and interpretability, its computational cost increases linearly with the number of active LoRA modules. This is a direct consequence of its design, which composes multiple LoRAs simultaneously in the latent space. In contrast, methods like Switch or Merge maintain a relatively constant cost by activating only a subset or a merged representation of LoRAs. This limitation aligns with prior observations in latent-space manipulation approaches (Zhong et al., 2024). The cost becomes particularly significant in dynamic settings, where all available LoRA adapters may be loaded concurrently. However, we also note that the MACs of LoRAtorio do not differ by orders of magnitude compared to previous works; in fact, we see that they are comparable, thus our method does not introduce a significant cost-performance trade-off compared to previous works. In Table 8b, we also see a comparison of LoRAtorio against reported inference latency in seconds. Our method has comparable latency to LoRA-composite and is significantly faster than MultLFG.

## I    QUALITATIVE COMPARISON

Examples comparing qualitatively our method against baselines for the SD1.5 base model can be seen in Figure 13 and Figure 14. LoRAtorio performs competitively, exhibiting fewer concept clashes and reduced vanishing of key attributes compared to previous SoTA. In the dynamic selection setting, we observe that Merge collapses, often producing non-legible or incoherent images, especially in the more complex reality subset. In contrast, LoRAtorio reliably selects the most relevant LoRA modules, resulting in sharper, more coherent generations, with clearly recognisable concepts.

Examples comparing qualitatively our method against baselines for Flux base model can be seen in Figure 15, Figure 16 and Figure 17. LoRAtorio shows lower concept confusion compared to merge. This is particularly obvious in the case of the Dynamic module selection setting, where the image quality severely deteriorates with multiple LoRAs. Even though Flux is a stronger model and thus generates more legible images compared to SD1.5 in the dynamic setting, the difference in quality compared to LoRAtorio and Naïve is substantial.

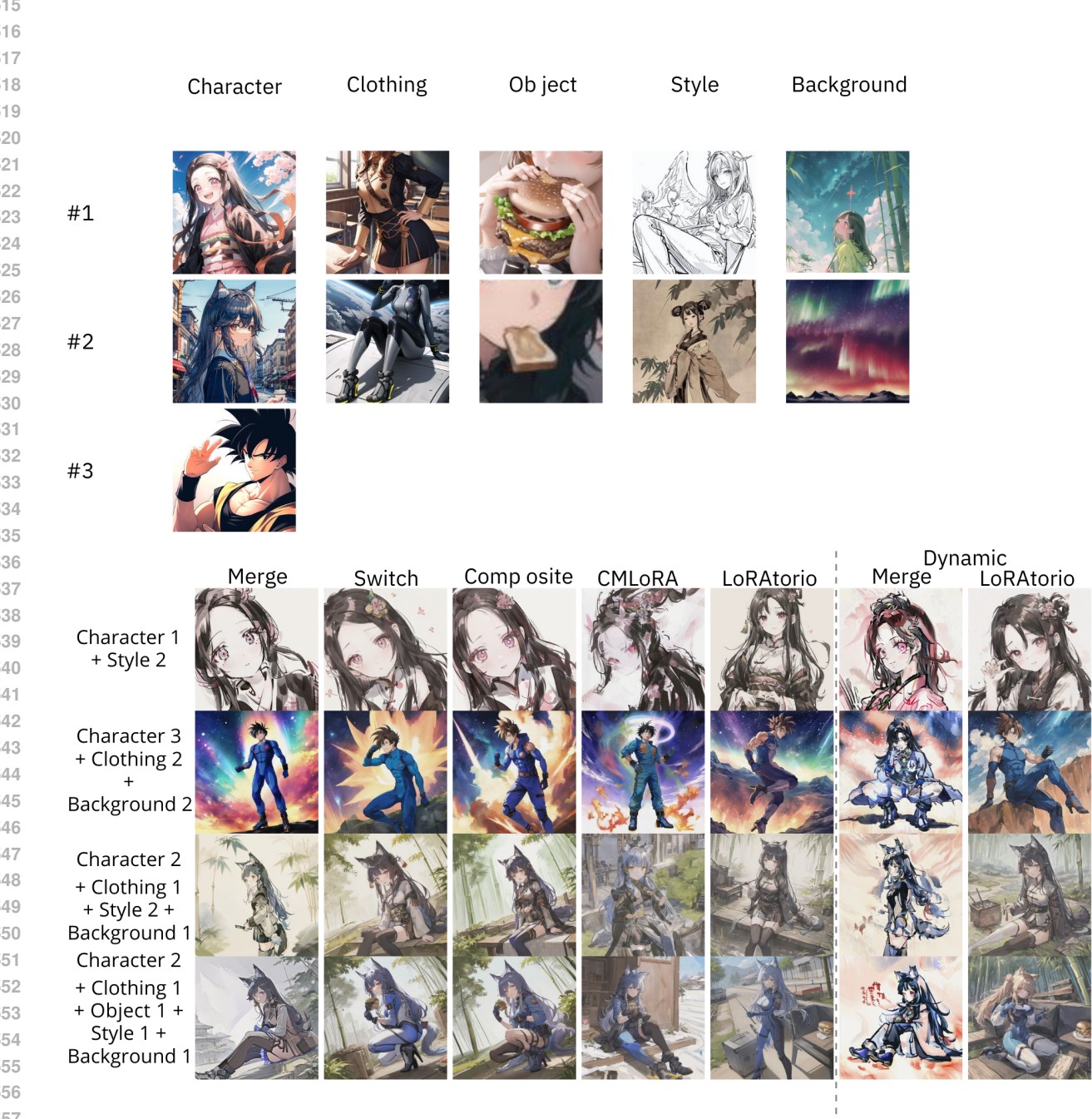

Figure 13: Images generated with $N$ LoRA candidates (L1 Character, L2 Clothing, L3 Style, L4 Background and L5 Object) across our proposed framework and baseline methods using SD1.5 base model on the anime subset of *ComposLoRA*.

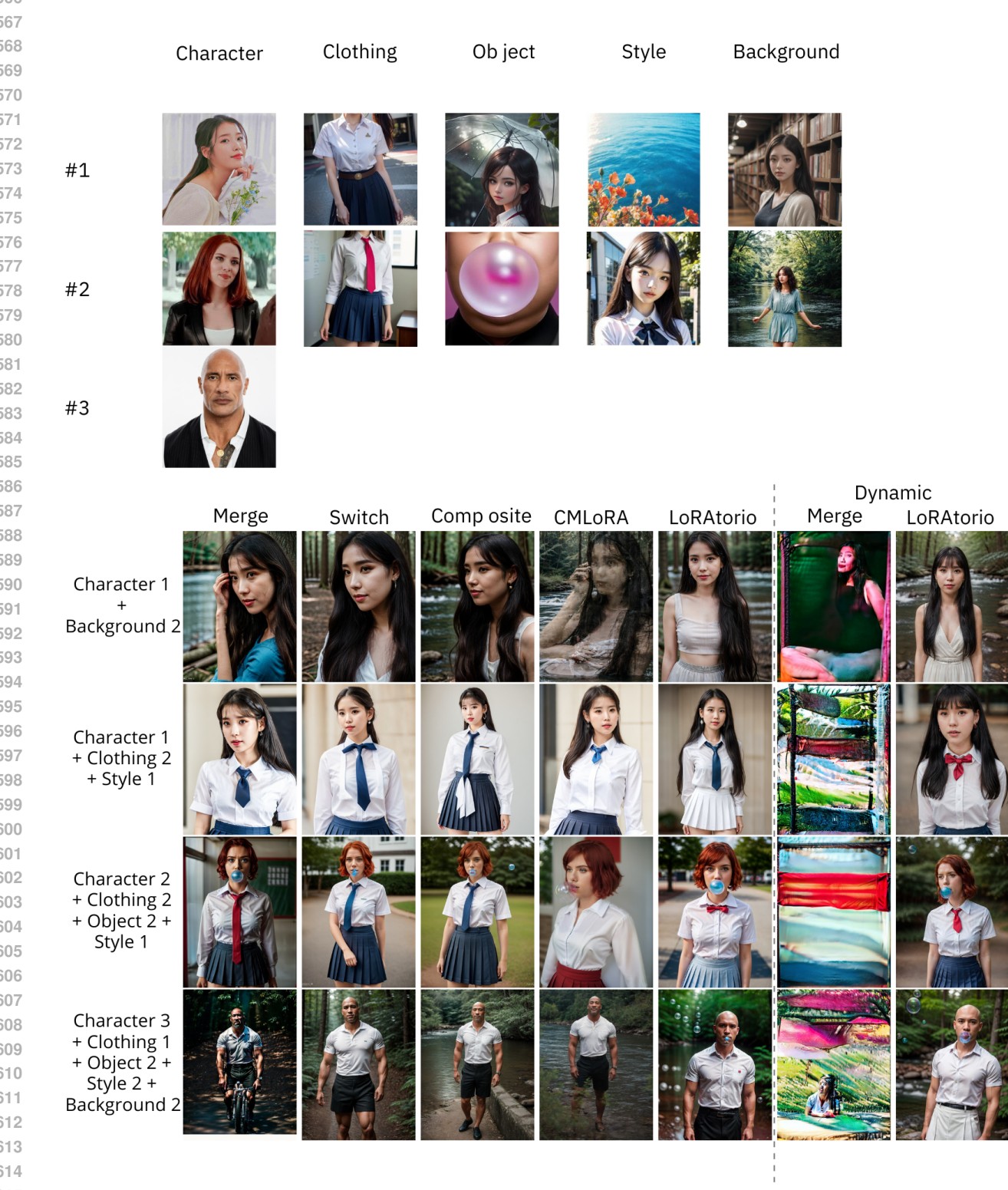

Figure 14: Images generated with $N$ LoRA candidates (L1 Character, L2 Clothing, L3 Style, L4 Background and L5 Object) across our proposed framework and baseline methods using SD1.5 base model on the reality subset of *ComposLoRA*.

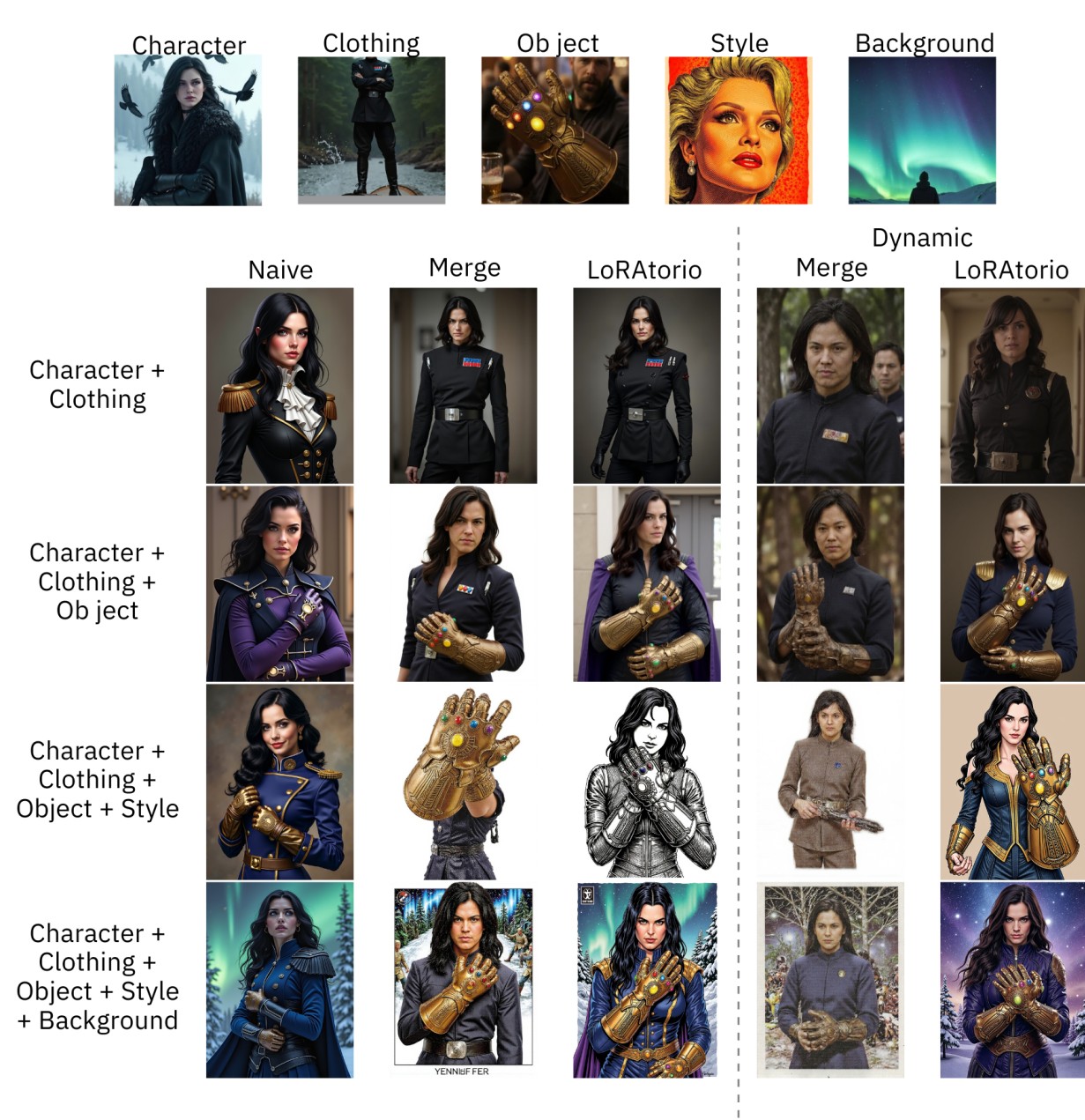

Figure 15: Images generated with $N$ LoRA candidates (L1 Character, L2 Clothing, L3 Style, L4 Background and L5 Object) across our proposed framework and baseline methods using Flux base model.

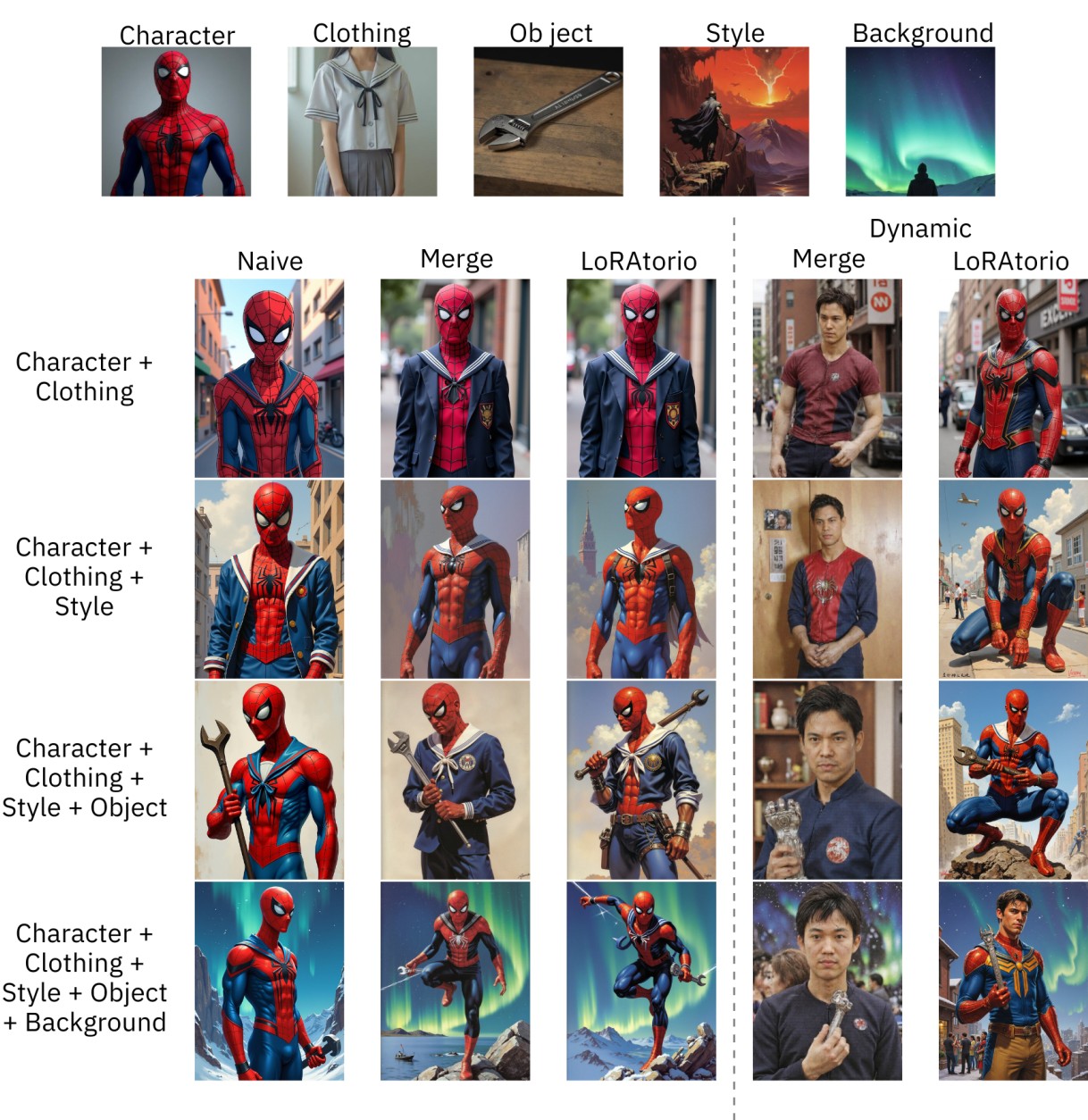

Figure 16: Images generated with $N$ LoRA candidates (L1 Character, L2 Clothing, L3 Style, L4 Background and L5 Object) across our proposed framework and baseline methods using Flux base model.

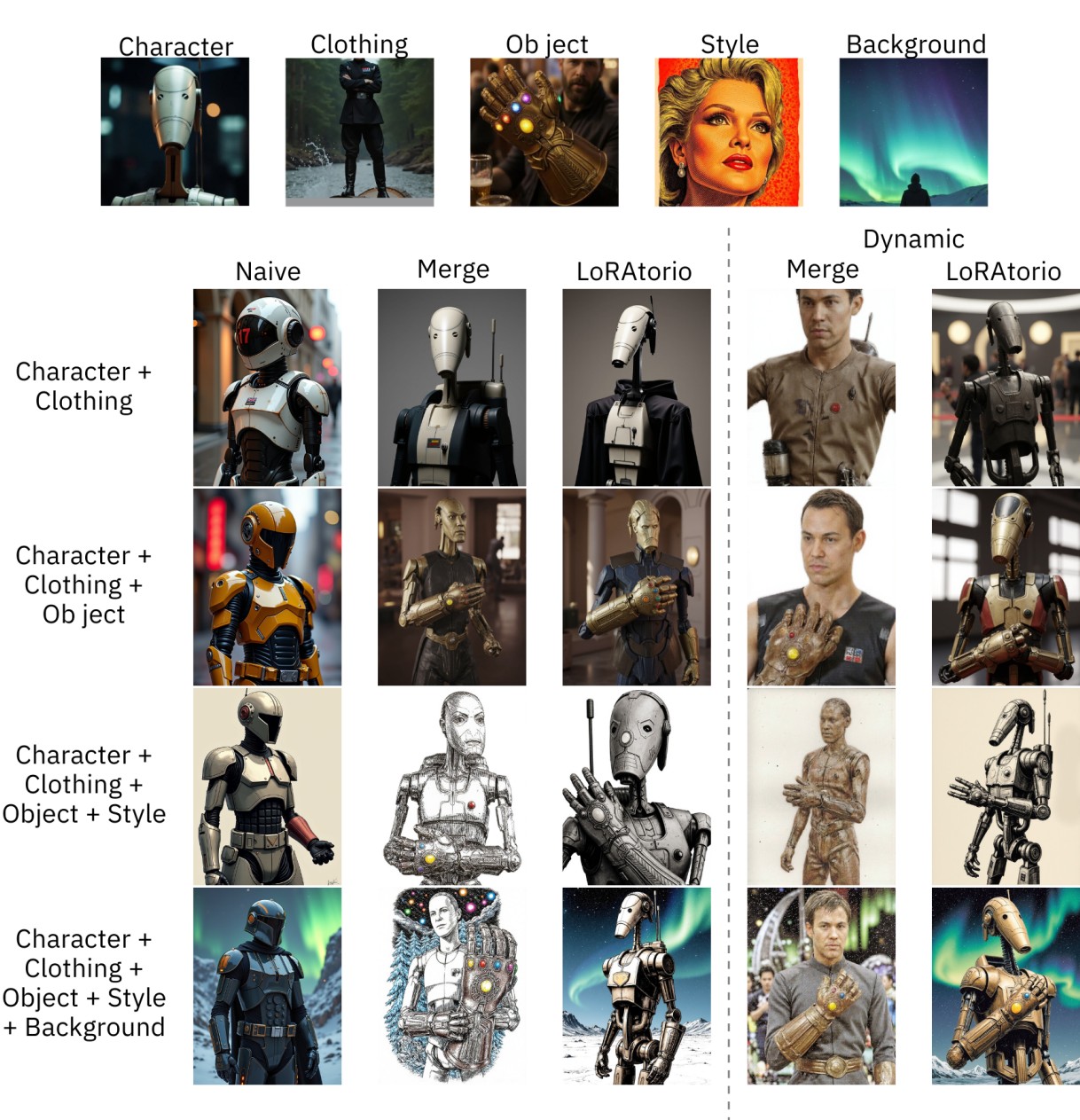

Figure 17: Images generated with $N$ LoRA candidates (L1 Character, L2 Clothing, L3 Style, L4 Background and L5 Object) across our proposed framework and baseline methods using Flux base model.

