# OpenReview forum: "LoRAtorio: An intrinsic approach to LoRA Skill Composition"
_ICLR.cc/2026/Conference — Submitted to ICLR 2026_

### Official Review · Reviewer_GGbc · 2025-10-22

**Soundness:** 2
**Presentation:** 2
**Contribution:** 2
**Rating:** 2
**Confidence:** 5

**Summary:**

The paper introduces LoRAtorio, a train-free framework for composing multiple LoRA adapters in text-to-image diffusion models. The method leverages intrinsic model behavior by computing patch-wise cosine similarity between LoRA outputs and the base model in latent space, using these similarities to weight contributions during denoising. It also proposes a modification to classifier-free guidance to mitigate domain drift and extends the approach to dynamic module selection for inference-time adaptability. Experiments on the ComposLoRA benchmark and Flux architecture show improvements in CLIPScore, GPT-4V evaluations, and human preference metrics compared to prior state-of-the-art methods.

**Strengths:**

Novelty: The idea of using intrinsic similarity for inference-time LoRA composition is original and avoids retraining, addressing practical constraints.

Comprehensive Evaluation: Includes automated metrics (CLIPScore), GPT-4V-based evaluation, and human studies across multiple datasets and architectures.

Dynamic Module Selection: Extends beyond static composition, which is relevant for real-world scenarios.

Clear Motivation: Observations about domain drift and latent divergence are well-supported by empirical analysis and visualizations.

Model-Agnostic Design: Demonstrates applicability to both Stable Diffusion and Flux architectures.

**Weaknesses:**

Computational Overhead: The method scales linearly with the number of LoRAs, making it impractical for large pools in dynamic settings. Latency (61–122s) is significantly higher than simpler baselines like Switch or Merge.

Limited Theoretical Depth: While cosine similarity is justified empirically, the theoretical motivation is relegated to an appendix and lacks rigorous formalism.

Evaluation Bias: Heavy reliance on CLIPScore and GPT-4V pairwise comparisons, which do not fully capture compositional fidelity or semantic correctness. Human evaluation is limited to three experts.

Failure Cases: The paper acknowledges severe failure modes (e.g., nonsensical outputs, concept confusion, duplicate limbs) but does not propose concrete mitigation strategies.

Assumption of LoRA Quality: The approach assumes LoRAs are well-trained and semantically coherent, which is unrealistic in community-driven repositories.

Scalability Concerns: Dynamic selection still requires loading all LoRAs into memory, which is infeasible for large-scale deployments.

Overstated Generalization: Claims of model-agnostic robustness are based on only two architectures; broader applicability remains unproven.

Ethical Section Superficiality: Mentions risks but lacks actionable guidelines or safeguards for misuse.

Generalizability: The methods have not been applied to large language models or multimodal large language models.

**Questions:**

How does LoRAtorio perform when LoRAs are trained on highly heterogeneous datasets with conflicting semantics?

Can the similarity-based weighting be approximated earlier in the pipeline to reduce computational cost?

How sensitive is the method to patch size and temperature hyperparameters in real-world scenarios?

Why was λ fixed at 0.5 for re-centering? Did you explore adaptive strategies?

Could metadata-driven pre-filtering or clustering of LoRAs improve dynamic selection efficiency?

How does the method handle cases where LoRA adapters introduce adversarial or biased features?

Is there any quantitative analysis of memory footprint for dynamic settings?

Would integrating learned gating (e.g., lightweight attention) outperform intrinsic similarity without full retraining?

---

> ### Author Response · Authors · 2025-11-14
>
> We appreciate Reviewer GGbc’s detailed review and acknowledgment of our contribution to improving subject fidelity, but several perceived weaknesses misrepresent the scope of the paper. For example, concerns about a lack of theoretical depth overlook the motivation formalised in Eq. 9–10. Regarding scalability, we refer the reviewer to the discussion in the Limitations and App. H, and urge a careful re-examination.
>
> Computational Overhead: We explicitly identify this as a weakness in the paper. Our aim, however, is subject fidelity not computational efficiency. As shown in App. H, our overhead is comparable to Composite and significantly lower than MultLFG.
>
> Theoretical depth: The motivation of our work is the mitigation of catastrophic forgetting already identified by works in LLMs and is formalised with eq. 9 and 10. We request clarification on what the reviewer considers “not rigorous” or insufficiently formal.
>
> Evaluation:  We follow established protocols from prior work. GPT-4V and human evaluations are aligned both here and in [1–3]. Works [1–2] also rely on the same number of human experts and are widely adopted. The metrics and protocol used are standard in this task and in image generation at large. If the reviewer requires criteria outside the domain’s norms, we request that they be explicitly listed and justified with references.
>
> Failure cases: We present these transparently as part of the method’s limitations. If mitigation strategies were known, they would be included. We believe explicit limitations benefit the community and should be viewed positively in review.
>
> Assumption of LoRA quality: LoRAs are publicly sourced via the ComposLoRA testbed. This assumption is inherent to the testbed itself. Our method addresses heterogeneity through the certainty proxy described in Method and formalised in App. A, achieving SOTA performance despite inevitable quality variability.
>
> Generalisation: UNet and DiT represent the two major diffusion backbones; Flux further introduces Rectified Flow. Covering two fundamentally different architectures provides stronger evidence of generality than evaluating within a single family. Extension to LLMs is completely out of the task scope, as autoregressive skill composition is fundamentally different from image generation.
>
> Ethical Section: The ethics section appropriately identifies and condemns misuse. Mitigation strategies constitute methodological advances and fall outside its intended purpose.
>
> Furthermore, to specifically answer the questions:
> - Heterogeneous datasets: This is already the setting of the ComposLoRA testbed. All reported results reflect exactly this scenario (see Related Work and Experiments).
>
> - computational cost: While App. A suggests this as a potential direction, efficiency is not addressed in this work. Limitations identify this explicitly as future work.
>
> - sensitivity: We would like to request a clarification of what constitutes a real-world scenario. We reiterate that all adapters used are sourced from the community website CivitAI, therefore, we believe that our method has already shown real-world in-the-wild capabilities.
>
> - λ: Fig. 8 shows λ’s qualitative effect, which is not captured by CLIPScore (see response to 2mHn). We choose λ = 0.5 for simplicity
>
> - metadata-driven: This is not something that we have explored in our work, as the current setting shows our patchwise similarity to work, even with the loaded LoRAs being in the dozens. Even if metadata-driven is possible, it is well outside the aims of the work (that is, to show distance from base model can serve as a proxy for confidence), and that is well counterintuitive given the motivation (that is, the metadata of publicly sourced adapters are rarely informative).
>
> - Bias: As concept disentanglement is not something the method aims to address, we do not explicitly handle adversarial or biased features. Given that we have no information on the training data or what adversarial or biased features would be in this context, it is impossible to measure such an effect and successfully mitigate it.
>
> - Memory: The dynamic setting scales linearly with the number of LoRAs, as noted in Sec. Limitations. For reference, generating an image takes 210 s with 22 loaded adapters. Our method addresses composition fidelity, not efficiency, and we deem the additional resources justified by the demonstrated improvements.
>
> - Retraining: We appreciate the suggestion but cannot retrain without access to training data which are unavailable for community-sourced adapters. Such an approach would fall under federated learning, which is a separate research problem.
>
> [1] X. Zou, et al, ‘Cached Multi-Lora Composition for Multi-Concept Image Generation’, ICLR 2025
>
> [2] Zhong et al. "Multi-LoRA Composition for Image Generation." TMLR 2024
>
> [3] Roy, A., et al. MultLFG: Training-free Multi-LoRA composition using Frequency-domain Guidance. preprint 2025

---

### Official Review · Reviewer_ftW8 · 2025-10-29

**Soundness:** 2
**Presentation:** 1
**Contribution:** 2
**Rating:** 2
**Confidence:** 4

**Summary:**

This paper presents LoRAtorio, a train-free framework for composing multiple LoRA adapters in text-to-image diffusion models. The authors identify key challenges in multi-LoRA composition—namely, semantic drift and performance degradation as more LoRAs are added—and address these by leveraging intrinsic model behavior rather than external supervision or retraining. Their approach partitions the latent space into spatial patches and computes cosine similarity between LoRA and base model noise predictions, constructing a spatially-aware weight matrix for adaptive skill fusion. They also propose a modification to classifier-free guidance to mitigate domain drift. Furthermore, LoRAtorio supports dynamic module selection at inference time, selecting relevant LoRAs on-the-fly. Experimental results on the ComposLoRA benchmark and additional settings demonstrate that LoRAtorio achieves better performance in both static and dynamic scenarios, outperforming prior works across automated metrics (CLIPScore), GPT-4V evaluations, and human assessments

**Strengths:**

The paper demonstrates originality by proposing a train-free, intrinsically guided framework for multi-LoRA composition, departing from the reliance on weight merging or learned gating. The quality of the work is evident in the methodology, including spatial patch-based weighting, re-centered guidance, and dynamic module selection. The paper is clearly written, with effective visualizations and thorough empirical support.

**Weaknesses:**

While the paper presents an innovative and effective approach, there are several notable weaknesses that merit attention. First, the authors do not release their code, which hinders reproducibility and weakens the reliability of the claimed results. Second, the core mechanism—spatial patch-based weighting—raises concerns when dealing with heterogeneous LoRA types. For example, style-oriented LoRAs may introduce global stylistic shifts across all spatial regions, while object-specific LoRAs affect only localized areas. The current similarity-based weighting may fail to harmonize such differences, potentially leading to outputs where object identity is distorted by the base model's style or vice versa, contrary to the intended composition. Third, a key advantage of LoRA is the ability to merge multiple adapters at inference with negligible overhead, but LoRAtorio requires evaluating all adapters independently at each step, increasing inference cost significantly. The paper lacks a discussion or analysis of this added complexity, which could impact its scalability in real-world applications. Addressing these issues would strengthen both the practicality and theoretical grounding of the work.

**Questions:**

1. Given the complexity of the method, open-sourcing the code would be essential for reproducibility and to support broader adoption in the community.

2. Have the authors evaluated or analyzed the behavior of their patch-based weighting mechanism when composing LoRAs of fundamentally different types—e.g., one encoding global stylistic shifts and another modeling localized objects? In such cases, a patch that diverges from the base model may not necessarily indicate higher relevance. Can the authors provide qualitative examples or ablations showing the composition quality in these mixed scenarios?

3. Relatedly, is there a risk that the current weighting method leads to mismatched compositions—e.g., an object generated with correct shape but styled according to the base model, while the background reflects the intended LoRA style? If so, how might this be mitigated?

4. The proposed method computes conditional and unconditional predictions for each LoRA independently at every denoising step, which appears to scale linearly with the number of adapters. Could the authors clarify the runtime and memory impact of their method compared to weight-merging baselines?

5. In the dynamic module setting, how well does the method scale when the number of candidate LoRAs is very large (e.g., dozens or hundreds)? Is the top-k selection based purely on per-step cosine similarity stable across timesteps, or is there a risk of inconsistency during denoising?

---

> ### Author Response · Authors · 2025-11-14
>
> We thank Reviewer ftW8 for highlighting the originality of our approach, the comprehensive evaluation, and the clear motivation supported by empirical analysis. However, most weaknesses identified are factually incorrect or overlook key sections of the paper; as such, we strongly urge the reviewer to read our paper and Appendices again, as most of the weaknesses and questions are already included in detail. More specifically, we clarify key misconceptions and errors in the review below:
>
> Code publishing: We have already committed to making the code public upon publication, as it can be clearly seen in the abstract of the paper. We urge the reviewer to read the abstract again.
>
> Composing LoRAs of fundamentally different types: The LoRA adapters in the ComposLoRA testbed used in this work are all heterogeneous. This is clear in the original paper introducing the testbed, but is also discussed in the Limitations of our paper. Regarding the relevance of global changes (eg, style) over local, our method takes a weighted average of the noise predictions using the similarity as weight (explained in detail in Sec 2.1), we expect the global information to still be retained in local patches. Furthermore, Figs. 12-16 showcase this in both the static and dynamic settings. We strongly urge the reviewer to revisit the method and qualitative results already provided and discussed.
>
> Risk of mismatched compositions: We once again want to clarify that, as discussed in Sec 2.1 and explicitly formalised in eq. 6, the method is not using the base model noise predictions in the denoising process; it is only using the divergence of noise prediction from the base prediction as a proxy for certainty and then uses this certainty measurement to take a weighted average. As such, and related to the previous explanation, we mitigate the risk of concept erasure or confusion. Please revisit Section 2.1 for a detailed explanation of the method.
>
> Runtime and memory impact: We already provide these details in Appendix H, Table 8.
>
> Dynamic module setting: The TopK gating system is indeed based on the cosine similarity, as made explicit in eq. 8. In Section 3.5, we show the choice of K in our setting, and Appendix I already shows how the dynamic selection performs in the dozens as all 22 adapters are concurrently loaded.
>
> We would also like to draw the attention of the AC to the fact that the review is entirely factually incorrect. Please assist us in making this discussion more productive.

---

> > ### Comment · Reviewer_ftW8 · 2025-11-14
> >
> > Thank you for the detailed response. I appreciate the effort the authors have taken to clarify various points.
> >
> > While I acknowledge the statement in the abstract that the code will be released upon publication, I want to clarify that my concern is not about whether the authors say they will release code, but rather about the verifiability of the current claims. The proposed method claims notable improvements over several strong baselines, yet its underlying assumptions are more heuristic than theoretically justified. Without access to the code or more principled reasoning, it is difficult to independently verify whether the performance gains genuinely stem from the proposed mechanism or from other factors (e.g., implementation details, hyperparameters, or evaluation setup). I suggest either making the code available during the review phase or providing a deeper analysis of why the method performs better than all compared baselines in both static and dynamic settings.
> >
> > The core of my concern is not simply whether the LoRAs used are heterogeneous, but whether the nature of their influence on the output space aligns with the proposed mechanism. Your method assumes that "patches where the LoRA strongly deviates from the base model reflect greater domain-specific influence"—however, this assumption does not necessarily hold in all cases. For instance, in stylistic LoRAs, the influence is often spatially uniform (affecting all regions to a similar degree), whereas in object-centric LoRAs, influence tends to be spatially localized. In cases where LoRA impact is globally homogeneous, the deviation signal used in your weighting may not carry meaningful localization, thus potentially failing to guide composition effectively. I raised the stylistic example precisely to illustrate this concern. If possible, I suggest including further analysis or failure cases when such assumptions do not hold.
> >
> > Risk of Mismatched Composition. This concern is a direct extension of the above. Even if the base model's noise is not directly injected into the denoising process, using deviation from base prediction as a proxy for confidence may introduce errors if the assumption about patch-local deviation does not align with the actual LoRA semantics. The weighting may underemphasize important contributions (e.g., style) or cause inconsistent blending across regions.
> >
> > Regarding runtime and memory: I notice the runtime measurements in Appendix H. To clarify, my request is not for additional experiments, but for a more explicit analysis of how computational cost scales with the number of LoRA adapters. A formal complexity characterization (e.g., O(NP) where N is number of adapters and P is number of patches, or similar) would help readers understand the practical implications and theoretical scalability of the method. Additionally, I did not find a discussion of memory consumption, which is an important aspect because LoRA’s advantage traditionally lies in its memory efficiency. Since LoRAtorio requires keeping all adapters active and computing similarity weights at each step, I would appreciate clarification or discussion on how memory usage scales in this framework.

---

> > > ### Author Response · Authors · 2025-11-18
> > >
> > > We thank the reviewer for the follow-up and for articulating the concerns with greater clarity. We address each point below and provide references to the corresponding sections, figures, equations, and appendices in the paper.
> > >
> > > Code availability and verification of claims: We agree that improved transparency is valuable. To support reproducibility *during* review, we note that:
> > > * Section 3.1 (Implementation Details) provides all hyperparameters, LoRA weights, guidance scales, patch sizes, temperature schedules, sampling methods, LoRA scaling factors, and hardware configuration. These are comparable to previous works, as clarified to the other reviewers, therefore all settings are ceteris paribus.
> > > * Appendix B.1 (Ablation Study) reports how performance changes when each component (patching, temperature, re-centering) is removed, helping isolate the contribution of each part of the mechanism. While some are purely qualitative (eg. re-centering), we note CLIPscore performance.
> > > * Appendix A (Theoretical Motivation) provides the underlying justification for similarity-based weighting, clarifying why the mechanism yields improvements.
> > > *  Figure 10 requested by reviewer 2mHn also visualises the patch-wise similarity to corroborate our claim.
> > > * We will add a short paragraph explicitly discussing which components most critically contribute to performance improvements in the main text, supporting the Appendices
> > >
> > > Behaviour with heterogeneous LoRAs (global style vs. local object):
> > > * The ComposLoRA testbed, which we use throughout (Section 3), explicitly contains *heterogeneous* LoRAs:
> > >   character LoRAs (object-centric), clothing LoRAs (local but structured), and style LoRAs (global).
> > > * Appendix C, Figures 9 and 10 show the temporal evolution and spatial maps of the similarity weights ( $\Omega_t$ ) when mixing *character* (local) and *style* (global) LoRAs.
> > > * Figure 10 in particular visualises the overlay of ( $\Omega_t$ ) for *Character + Style*. We observe:
> > >   * Character LoRA receives high weights on localised patches.
> > >   * However, we also see that between steps style LoRA also has an effect, therefore global concepts are not locally nullified.
> > > * Section 2.1 (Skill Composition Using Intrinsic Knowledge), especially the paragraph beginning *“We interpret cosine similarity in the noise prediction…”*, explains that similarity is interpreted as a relevance proxy, not strictly spatial locality.
> > > * In the global-style case:
> > >  * The averaged per-patch similarity in Eq. (3)–(4) yields a consistent but lower similarity across all patches.
> > >  * After SoftMin, style LoRA receives a consistent—but not overpowering—weight across all spatial regions.
> > > Risk of mismatched composition (object shape vs. style transfer):
> > > * Section 2.1, Eq. (6) formalises that the final conditional score is a weighted sum of *LoRA conditional predictions only*. The base model appears only as a reference point for similarity.
> > > * Re-centering prevents the unconditional term from pulling toward end of distribution style or semantics. Figure 4 and Appendix B.1 (Figure 8) empirically demonstrate that improper re-centering (e.g., $\lambda=0$ or $\lambda=1$) leads to precisely the mismatch the reviewer is concerned about—over-stylisation or loss of identity
> > > * Figures 12–16 (in the main PDF’s qualitative appendices) show compositions where different LoRAs are blended without mismatched textures or base-model bleed-through.
> > > * Therefore we note, that we have already both theoretically and empirically covered this in the paper.
> > >
> > > Computational complexity and scaling with number of LoRAs:
> > > * As mentioned in Appendix H, the complexity scales linearly with each additional LoRA which translates to O(N). Then it is indeed O(N+P) for the number of patches. We will add this O notation to the camera-ready.
> > > * Each LoRA adapter adds only its rank-r matrices, not a full duplicate of the model; this is not our contribution, but rather an innate property of the adapters themselves. We already include memory details in Appendix H, however we will clarify that these do not fluctuate per step.
> > >
> > > Stability of Top-K selection in dynamic setting:
> > > * Appendix C (Temporal Analysis), including Figure 9, shows the evolution of average ( \Omega_t ) across timesteps in the simple 2 LoRA case.
> > > * Empirically, Figures 12–16 show that dynamic top-k yields stable composition with no significant artefacts or concepts and styles from unwanted LoRAs. In addition, we see that ClipScores do not significantly drop in the dynamic case, thus minimal concept deletion or confusion occurs.

---

### Official Review · Reviewer_2mHn · 2025-11-02

**Soundness:** 2
**Presentation:** 3
**Contribution:** 1
**Rating:** 4
**Confidence:** 4

**Summary:**

LoRAtorio is a train-free framework for composing multiple LoRA adapters in text-to-image diffusion models. It leverages the observation that LoRAs deviate from the base model on in-distribution inputs but remain close on out-of-distribution ones, using cosine similarity between LoRA and base outputs in latent space to guide spatially weighted aggregation. The method also introduces a modified classifier-free guidance term to mitigate domain drift and supports dynamic, inference-time selection of relevant adapters. LoRAtorio achieves state-of-the-art performance, improving CLIPScore and human-evaluated visual quality across multiple diffusion architectures.

**Strengths:**

1. The authors propose spatially-aware similarity metric to use as a proxy for LoRA adapter's confidence, with sound theoretical motivation.
2. The authors extend the task of multi-LoRA composition to a dynamic module selection setting, which is a good, real-world skill composition scenario.

**Weaknesses:**

1. The first contribution seems to be incremental - MultLFG (2nd best method) proposes "... training-free frequency-aware multi-LoRA merging. The key idea is to decompose LoRA-based noise predictions into frequency subbands and perform adaptive merging based on relevance scores." (https://arxiv.org/pdf/2505.20525), whereas this paper proposes patched cosine distance instead of frequency subbands.
2. The second contribution - re-centering - is, per your results in Table 6a, only better by 0.01 (36.543 with vs 36.532 w/o) CLIPScore, on a limited ablation study (see Weakness 3), which makes me believe it does not improve anything. What are the standard deviations for these results?
3. You compare your method to 8 reference methods in Table 1, just 4 in Table 2 and only 2 in the rest (Table 3, 4, 5c/d). The 2nd best performing method (per Table 1), MultLFG, is only shown once and never mentioned again. Why is that?
4. Ablation study of the proposed method (Table 6) lacks the same detail as, for example, Table 1. It only analyzes 2 or 3 component scenarios (missing 4 and 5), only on one subset (anime), on one backbone (stable-diffusion-v1.5).

**Questions:**

1. Could you extend the analysis of the weight matrix Ω? For example, I am wondering if taking the most dominant adapter's category per each patch could result in semantic masks appearing. Especially non global categories like character or object may be visible.

---

> ### Author Response · Authors · 2025-11-14
>
> We appreciate Reviewer 2mHn’s acknowledgement of the sound theoretical motivation behind our spatially-aware similarity metric and the practical relevance of dynamic module selection for real-world skill composition. Your comments on comparative analysis and ablation depth are valuable, and we address these points in detail below.
>
> Comparison with MultLFG: We respectfully, fundamentally disagree with the reviewer and urge him to revisit our method. Indeed, MultLFG "decompose LoRA-based noise predictions into frequency subbands and perform adaptive merging based on relevance scores". This refers to the frequency of the denoised, RGB image. Our method does not look into frequency; instead, we use the predicted noise and compare it to the predicted noise of the base model as a measure of certainty. These methods are not only conceptually fundamentally different in what they measure (RGB frequency vs noise), but also in the space that they operate and the resources they require (RGB vs latent). As such, our method is far from incremental to MultLFG, we propose a fundamentally different approach using intrinsic model properties rather RGB space features. We hope that revisiting the method section and the additional explanation given here clarifies the misunderstanding. We also update the related work (lines 456-462) to explicitly explain the difference in approaches.
>
> Contribution of re-centering: As the main contribution of re-centering is subject fidelity, we want to reiterate what we have discussed in the results and what previous works have identified: that is, CLIPScore is somewhat limited, due to language ambiguity and modality gaps. As such, we would like to refer the reviewer to Fig. 8 in the appendix, where the qualitative impact of re-centering is visible and where we discuss the need for empirical choice of lambda rather than ClipScore reliance. In practice, a lambda of 1 would translate to no re-centering applied; with no re-centering, the composition remains the same, however, important instance elements (eg, the gold top, the hair of the character, or the shoes) all lack visual details that are essential to visually recognise the concepts but cannot be captured by CLIPScore (for reference all images in the first row of Fig. 8 including the autoguidance sample, have CLIPScore 32.5519). In short, we believe the CLIPScore to be a good metric when it comes to identifying whether the object/concept is present, but not to judge instance fidelity, for which qualitative evaluations are necessary. We have updated Fig. 8 and text in lines 806-811 to explicitly show CLIPScore capturing elements' presence but not the instance fidelity or image quality, and the need for empirical guidance.
>
> Comparison with MultLFG: This work does not have public code or evaluation images; therefore, any qualitative and pairwise comparison is impossible. In addition, we want to mention that MultLFG is not published; we felt it was important to include the results, as they are indeed the second best, but have not been through a peer review process at the time of the submission and have no public code and images for evaluation.
>
> Ablations: Typically, ablations are done on one dataset and not the entire evaluation testbed. In addition, given the complementary nature of CLIPScore to qualitative aspects, we are not sure if extensive ablations on a single metric are representative; however,  we will extend the ablation to include more N in the following days and for the camera-ready.
>
> Analysis on Omega: We welcome the suggestion. In addition to Fig. 9, which already shows the Omega for the entire image, we update the paper with Fig. 10, which shows the upscaled Omega of the latent as a heatmap over the RGB image.

---

### Official Review · Reviewer_rah2 · 2025-11-02

**Soundness:** 3
**Presentation:** 3
**Contribution:** 3
**Rating:** 4
**Confidence:** 4

**Summary:**

The paper presents LoRAtoria, a novel train-free framework for multi-LoRA composition that leverages intrinsic model behavior.
The framework consists of two parts: skill composition on the patch level and re-centering of the unconditional noise output.
Also, the paper introduces MultiLoRA composition task with a dynamic LoRA selection

**Strengths:**

1. The paper is well structured and easy to follow.
2. The proposed approach demonstrates better results with increasing the active LoRA adapters
3. Re-centering of the unconditional noise could be used independently
4. Both UNet and DiT-based models are checked
5. The human and VLM-based evaluations are fully described
6. Extensive appendix

**Weaknesses:**

1. MultiLoRA composition task with a dynamic LoRA selection probably requires more detailed description as now it lacks motivation (at least some potential use cases)
2.The majority of the comparisons are done using CLIPScore that is a good proxy metric; however, a more extensive human or VLM-based evaluation is suggested
3. Only composition of LoRas for the Character, Style and Background are considered. No compositions with LoRAs for faster inference (e.g., LCM) are checked
4. see questions

**Questions:**

1) inconsistent d:
* lines 213-214: "$d$ is the number of pixels per patch"
* lines 226-227 mention upscaling to $H/d \times W/d$
Please use $d^2$ as the number of pixels in patch or upscaling to $\sqrt(d)$ in the blocks description
2) the commonly used number of diffusion steps for SD1.5 is 30-50 steps (towards 30 if DPM++ solver is used); however, in the section 3.1 authors mentioned 100 steps for realistic subset and 200 steps for the anime subset without any further explanation.
3) The human evaluation mentioned only 3 experts. Have you considered running the quality assessment using GPT4v?
4) The results for the Rectified Flow are presented only on FLUX.1-dev checkpoints that is guidance distilled (re-centering couldn't be applied) while Stable Diffusion 3.5 is not checked.
5) Comparison with AutoGuidance is presented in the appendix; however, AutoLoRA(https://arxiv.org/abs/2410.03941) shows that AutoGuidance-ish approach could be combined with CFG for LoRAs
6) I believe that the skill composition process could be better illustrated

---

> ### Author Response · Authors · 2025-11-14
>
> We thank Reviewer rah2 for noting the clarity and structure of the paper, as well as the robustness of our evaluation across UNet and DiT-based models. We appreciate your recognition of the extensive appendix and the detailed human and VLM-based evaluations. Below, we address your suggestions regarding motivation, evaluation scope, and technical clarifications.
>
> Motivation of dynamic Multi-LoRA composition task: We welcome the suggestion and update the introduction (lines 115-120) to better explain the motivation of the dynamic LoRA selection task.
>
> Human and VLLM evaluation: We strongly urge the reviewer to revisit the paper. Section 3.3 of the main paper presents results using GPT-4V, and Section 3.4 provides supporting qualitative human analysis. In addition, Appendix B provides a deep dive into these results, and Appendices F and G provide the code and the interface used to obtain them.
>
> LCM: We are aware of LCM adapters, however, such types of adapters are outside the remit of the task, as ultimately it is related to image generation content and not inference speed. Furthermore, in the MultiLoRA composition task, the adapters are trained in DreamBooth style and are triggered by specific text in the generation, which allows for Observations 1 and 2. As such, any transfer to a different task can only be examined in future work, as we have no insight into how LoRAs for faster inference behave in the latent space or if differences can be observed.
>
> Use of d: we have updated line 213 for clarity and consistency.
>
> Number of steps: We clarify that this is the same number of steps used by previous works [1,2,3], and even though they are not typical in other image generation tasks, they are the norm in the ComposLoRA testbed.
>
> Flux: We chose Flux over StableDiffusion-3.5 as the aim was to show the composition method can be applied to multiple families of denoising networks, as identified in the strengths of this review. We will add samples of images generated with SD3.5 and our method in the camera-ready.
>
> Auto-guidance: We thank the reviewer for the opportunity to discuss this aspect. The original idea to improve fidelity does indeed stem from the AutoGuidance paper. However, as we can see from the qualitative output in Fig.8 of the appendix, the generated images appear to be from the extremes of the distribution. While AutoLoRA achieves high-quality in the single LoRA scenario, the fundamental difference in our setting is the integration of multiple adapters that significantly affect the data distribution, as shown in Fig.1 and discussed briefly in that section. Ultimately, the different experimental setup makes the two otherwise similar methods (AutoLoRA and our re-centering approach) not comparable.
>
> Illustration: We welcome any suggestions the reviewer has.
>
> [1] X. Zou, M. Shen, C.-S. Bouganis, and Y. Zhao, ‘Cached Multi-Lora Composition for Multi-Concept Image Generation’, ICLR 2025
>
> [2] Zhong, Ming, Shuohang Wang, Yadong Lu, Yizhu Jiao, Siru Ouyang, Donghan Yu, Jiawei Han, and Weizhu Chen. "Multi-LoRA Composition for Image Generation." TMLR 2024
>
> [3] Roy, A., Suin, M., Shah, K. and Chellappa, R., 2025. MultLFG: Training-free Multi-LoRA composition using Frequency-domain Guidance. preprint 2025

---

### Author Response · Authors · 2025-11-14
**Summary of changes**

We thank all reviewers for their time and constructive feedback. We are encouraged by the recognition of our paper’s originality in proposing a train-free, intrinsically guided framework for multi-LoRA composition (ftW8), its clear structure and comprehensive evaluation across automated metrics, GPT-4V, and human studies (rah2, ftW8), and its applicability to both Stable Diffusion and Flux architectures (rah2, ftW8). Reviewers also highlighted the sound theoretical motivation behind our spatially-aware similarity metric (2mHn) and the practical relevance of dynamic module selection for real-world scenarios (2mHn).

To address some of the reviewers' concerns, we have updated the submitted manuscript with the following:
- Lines 115-120: We update to provide a motivation for the dynamic extension of the task (rah2)
- Line 213: We correct the patch size to d^2 (rah2)
- Lines 456-462: We provide clarification on key differences with frequency-based approaches (2mHn)
- Fig 8 caption and lines 806-811: clarify the use of lambda and how it affects image quality. (2mHn)
- Fig 10: Omega map for character visualised (2mHn)

Detailed responses are added to each reviewer’s comments.

---

### Author Response · Authors · 2025-11-28

Dear reviewers,

Thank you for your time and constructive feedback. As the discussion phase ends in less than one week, we would like to confirm whether our previous responses have adequately addressed your concerns. If any points still require clarification, we would greatly appreciate hearing from you soon so we can respond within the remaining discussion window.

---

### Author Response · Authors · 2025-12-02

Given the OpenReview incident, we would like to briefly summarise the key points discussed before the rebuttal was interrupted.

Reviewer **rah2** highlighted the clear structure of the paper, the robustness of our evaluation with increasing number of LoRA adapters, and the applicability of our method to both UNet- and DiT-based architectures (Stable Diffusion and Flux), while suggesting a clearer motivation for the dynamic Multi-LoRA setting, and pointing out minor inconsistencies in notation; we addressed these by updating the introduction to better motivate the dynamic task (lines 115–120), fixing the definition of the patch size (d^2) (line 213). In addition, we clarified experimental details that are standard in the task, the use of GPT4V for large scale qualitative evaluation that was already included in the paper and discussed comparison with different works that nevertheless operate in the single LoRA case.

Reviewer **2mHn** recognised the theoretical motivation of our spatially-aware similarity metric and the practical relevance of the dynamic module-selection extension, but raised concerns about perceived incremental similarity to MultLFG, the contribution of re-centering, and the breadth of ablations. In response, we clarified in the related work (lines 456–462) that MultLFG operates in RGB frequency space whereas our approach is an intrinsic proxy to model uncertainty operating in the noise/latent space, expanded the qualitative discussion and Fig. 8 (lines 806–811) to better illustrate the impact of re-centering, and added Fig. 10 to visualise the Ω maps and strengthen interpretability.

Reviews **ftW8** and **GGbc** acknowledged the originality of our train-free, intrinsic framework, the comprehensiveness of our evaluation, and the relevance of dynamic selection, while raising several concerns. Many of these points appeared to arise from differences in interpretation of the scope and goals of the work or from material already included in the submission (e.g., code release commitment, runtime and memory analysis, and formal motivation, all covered in the abstract, Sec. 3.1, Eq. 9–10, and Apps. A, H, I). We clarified these aspects with detailed pointers and appreciate the opportunity to make the relevant sections more explicit.

Although the reviewers were unable to respond due to the incident, we remain cautiously confident that our responses would have addressed their concerns.

---

### Meta-Review · Area_Chair_zRG8 · 2026-01-04

**Summary:**

The paper proposes LoRAtorio for train-free composition and dynamic selection of multiple LoRA skills. Reviews are negative: two 4, two 2, with key concerns around verifiability, scalability/latency, and the strength of the evidence for the proposed “global style vs. local object” assumption. Additionally, the submission does not use the official ICLR template, which raises a compliance/fairness issue and should be flagged.

**Reviewer Concerns:**

Addressed (partly) in rebuttal: clarified notation and implementation details (e.g., patch size, steps), better articulated differences to prior Multi-LoRA baselines, added qualitative explanations, and discussed runtime/complexity scaling with the number of LoRAs.

Still outstanding: limited failure-case/robustness analysis for the core assumptions; practical overhead for large LoRA pools remains a concern; evaluation reliance on CLIPScore/GPT-4V with limited human validation is only partially de-risked; and the non-ICLR template issue remains unresolved (needs PC/SAC handling).

**Reviewer Scores:**

rah2 (4): likely 4 → 5 (clarifications help).
2mHn (4): likely 4 → 4 (still wants stronger evidence/ablations).
ftW8 (2): likely 2 → 2/3 (core concerns likely remain).
GGbc (2): likely 2 → 2 (unlikely to change).

---

### Decision · Program_Chairs · 2026-01-26

Reject